# Robust Plug-and-Play Joint Axis Estimation Using Inertial Sensors

**DOI:** 10.3390/s20123534

**Published:** 2020-06-22

**Authors:** Fredrik Olsson, Manon Kok, Thomas Seel, Kjartan Halvorsen

**Affiliations:** 1Systems and Control, Department of Information Technology, Uppsala University, SE-75105 Uppsala, Sweden; kjartan.halvorsen@it.uu.se; 2Delft Center for Systems and Control, Delft University of Technology, 2628 CD Delft, The Netherlands; m.kok-1@tudelft.nl; 3Control Systems Group, Technische Universität Berlin, 10623 Berlin, Germany; seel@control.tu-berlin.de; 4Department of Mechatronics, Campus Estado de Mexico, Tecnologico de Monterrey, Monterrey 64849, NL, Mexico

**Keywords:** inertial measurement units, gyroscopes and accelerometers, sensor-to-segment calibration, kinematic constraints, joint axis identification, validation on mechanical joint

## Abstract

Inertial motion capture relies on accurate sensor-to-segment calibration. When two segments are connected by a hinge joint, for example in human knee or finger joints as well as in many robotic limbs, then the joint axis vector must be identified in the intrinsic sensor coordinate systems. Methods for estimating the joint axis using accelerations and angular rates of arbitrary motion have been proposed, but the user must perform sufficiently informative motion in a predefined initial time window to accomplish complete identifiability. Another drawback of state of the art methods is that the user has no way of knowing if the calibration was successful or not. To achieve plug-and-play calibration, it is therefore important that 1) sufficiently informative data can be extracted even if large portions of the data set consist of non-informative motions, and 2) the user knows when the calibration has reached a sufficient level of accuracy. In the current paper, we propose a novel method that achieves both of these goals. The method combines acceleration- and angular rate information and finds a globally optimal estimate of the joint axis. Methods for sample selection, that overcome the limitation of a dedicated initial calibration time window, are proposed. The sample selection allows estimation to be performed using only a small subset of samples from a larger data set as it deselects non-informative and redundant measurements. Finally, an uncertainty quantification method that assures validity of the estimated joint axis parameters, is proposed. Experimental validation of the method is provided using a mechanical joint performing a large range of motions. Angular errors in the order of $2^{\circ }$ were achieved using 125–1000 selected samples. The proposed method is the first truly plug-and-play method that overcome the need for a specific calibration phase and, regardless of the user’s motions, it provides an accurate estimate of the joint axis as soon as possible.

## 1. Introduction

Wearable inertial measurement units (IMUs) have become a key technology for a range of applications, from performance assessment and optimization in sports [1], to objective measurements and progress monitoring in health care [2], as well as real-time motion tracking for feedback-controlled robotic or neuroprosthetic systems [3]. In all these application domains, IMUs are used to track or capture the motion of mechatronic or biological joint systems such as robotic or human limbs. In this work we consider such systems where the joint is a hinge joint with one degree of freedom. Examples of hinge joints include the knee and finger joints, which are essential in applications targeting lower limb [4] and hand [5] kinematics.

In contrast to stationary optical motion tracking systems, miniature IMU networks can be used in ambulatory settings and facilitate motion tracking outside lab environments. While this is an important step towards ubiquitous sensing, one major limitation of the technology is that the IMUs’ local coordinate systems must be aligned with the anatomical axes of the joints and body segments to which they are attached. This sensor-to-segment calibration is a crucial step that establishes the connection between the motion of the IMUs and the motion of the joint system to which they are attached.

Several different approaches have been proposed for sensor-to-segment calibration of inertial sensor networks, from trying to align the sensor axes with body axes by precise attachment to predefined calibration poses and motions; see, e.g., [6,7,8,9]. However, in all of these cases, the calibration crucially depends on the knowledge and skills of the person who attaches the sensors or the person who performs the calibration procedure. This might be acceptable in supervised settings with trained and able-bodied users, but it represents a major limitation of IMU-based motion tracking and capture in clinical applications and in motion assessment of elderly and children. Finding solutions for these application domains and enabling ubiquitous sensing in daily life requires the development of less restrictive methods for sensor-to-segment calibration.

Ideally, wearable IMU networks should be plug-and-play, and the sensor-to-segment calibration should be performed by the network autonomously, which means without additional effort or requirements on the user’s knowledge or on the performed motion. An important step towards this goal was the development of methods that exploit the kinematic constraints of the joints to identify sensor-to-segment calibration parameters from almost arbitrary motions [10,11]. For joints with one degree of freedom (DOF), the feasibility of this approach has been demonstrated [12,13,14,15]. Methods have been proposed that require the user to perform a sufficiently informative but otherwise arbitrary motion during an initial calibration time window and determine the functional joint axis in intrinsic coordinates of both IMUs, cf. Figure 1. It was recently shown that almost every motion, including purely sequential motions and simultaneous planar motions, is informative enough to render the joint axis identifiable unless the joint remains stiff throughout the motion [16].

Several methods targeting different types of joints or sensor-to-segment calibration parameters have been developed. In [17], a method for identifying the joint axes of a joint with two DOF was proposed. Methods for identifying the position of the joint center relative to sensors attached to adjacent segments have been proposed in [12,18,19]. A method enabling automatic pairing of sensors to lower limb segments have been proposed in [20].

The published kinematic-constraint-based methods constitute an important step forward but still impose undesirable and unnecessary limitations. If the user does not move during the initial calibration time window or if the motion is not sufficiently informative, the calibration will be wrong and all subsequently derived motion parameters will be subject to unpredictable errors. For a truly plug-and-play system, it is therefore crucial that the IMU network is able to
Recognize how informative motions are and whether they render the joint axis identifiable;Wait for sufficiently informative data to be generated and combine useful data even if it is spread and intermitted by useless data;Determine how accurate the current estimate of the joint axis is and provide only sufficiently reliable estimates.

An IMU network with such properties can be used without the aforementioned limitations. Once it is installed, it will autonomously gather all available useful information and provide reliable calibration parameters as soon as possible, which immediately enable calculation of accurate motion parameters from the incoming raw data as well as from already recorded data. To explain the practical value of the proposed concept of plug-and-play calibration, we briefly compare this concept to the aforementioned existing calibration concepts that use predefined motions [6,7,8,9] or arbitrary motions [10,11,12,13,14,15]:

*Predefined-Motions:* The calibration is based on the assumption that the user performs a sequence of predefined motions and poses with sufficient precision within a predefined initial time interval. The approach fails and provides inaccurate calibration without warning if
(a)The user performs the sequence of predefined motions and poses without sufficient precision;(b)The user performs the sequence with sufficient precision but not within the predefined initial time interval;(c)The user performs sufficiently informative but otherwise arbitrary motions;(d)The user performs no sufficiently informative motion at all, e.g., he/she moves with a stiff joint.

*Arbitrary-Motions:* The calibration is based on the assumption that the user performs sufficiently informative but otherwise arbitrary motions within a predefined initial time interval. The motion does not need to be precise, and it has been shown that sufficient excitation is provided by almost every motion for which the joint does not remain stiff [16]. However, the approach fails and provides inaccurate calibration without warning if
(a)The user performs a sequence of predefined motions but not within the predefined initial time interval;(b)The user performs sufficiently informative arbitrary motions but not within the predefined initial time interval;(c)The user performs no sufficiently informative motion at all, e.g., he/she moves with a stiff joint.

*Plug-and-Play:* The proposed sensor-to-segment calibration approach. It works well for all mentioned cases and exceptions in the sense that it always provides accurate calibration parameters as soon as the user’s motions are sufficiently informative, and it clearly indicates at all times whether the desired calibration accuracy has yet been reached.

It is important to note that the cases without warning are very dangerous, because inaccurate information is provided and claimed as accurate. In many applications, this leads to unacceptable risks. This and the other listed differences between the two existing approaches and the proposed new method have large implications for the way wearable IMU networks can be used in offline and online applications.

*Offline Applications* include motion capture for ergonomic workplace assessment [21], for monitoring of movement disorders [2] and for sport performance analysis [1]. In state-of-the-art solutions, the user performs an initial calibration procedure before (or after) recording data from the motions to be analyzed. The user can only hope that the calibration was accurate enough. If the calibration was inaccurate, then all recorded data is corrupted and might lead to false interpretation and conclusions. In contrast, when the calibration is plug-and-play, the user starts recording data from motions that should be analyzed immediately after attaching the sensors. Calibration automatically takes place as soon as sufficiently informative data has been gathered. The system indicates that calibration has been successful, and the user can be sure that all obtained measurements are valid and accurate. The identified calibration parameters are used to evaluate the data that was recorded before and after the moment at which accurate calibration was achieved.

*Online Applications* include real-time motion tracking for wearable biofeedback systems [22] as well as robotic and neuroprosthetic motion support systems [23]. In state-of-the-art solutions, the user first performs an initial calibration procedure before the sensor system is connected to an assistive device that uses the measurements to provide e.g., biofeedback or motion support. The user can only hope that the calibration was accurate enough. If the calibration was inaccurate, then the provided biofeedback or motion support might be wrong and dangerous. In contrast, when the calibration is plug-and-play, the user instead attaches the sensors and starts moving. As soon as the desired calibration accuracy has been achieved, the sensor system automatically provides measurements to the assistive device. The user can be sure that all provided biofeedback and motion support is based on valid and accurate measurements.

In the present contribution we propose the first joint axis identification method for one-dimensional joints that is plug-and-play in the aforementioned sense. The main contributions of the present work are the following:We leverage recent results on joint axis identifiability [16] to develop a sample selection method that overcomes the limitation of a dedicated initial calibration time window.To assure that the motion needs to fulfill only the minimum required conditions, we combine accelerometer-based and gyroscope-based joint constraints and weight them according to the information contained in both signals.We propose an uncertainty quantification method that assures validity of the estimated joint axis parameters and thereby eradicates the risk of false calibration.We provide an experimental validation in a mechanical joint performing a large range of different motions with different identifiability properties.

In the proposed system, successful calibration no longer depends on performing certain motions in a predefined manner or time window but only on fulfilling the minimum required conditions at some point. Moreover, the system knows when these conditions are fulfilled and provides only reliable calibration parameters.

## 2. Inertial Measurement Models

Inertial sensors collectively refers to accelerometers and gyroscopes, which are sensors used to measure linear acceleration and angular velocity, respectively. When the sensors have three sensitive axes which are orthogonal to each other, the inertial sensors can measure these quantities in three dimensions. Such sensors are referred to as triaxial. An IMU is a single sensor that contains one triaxial accelerometer and one triaxial gyroscope. The measurements from the IMU are obtained with respect to (w.r.t.) a reference frame, referred to as the sensor frame (S), its axes and origin corresponding to those of the accelerometer triad. The axes of the gyroscope is assumed to be aligned with the axes of the accelerometer. The measured quantities describe the motion of the sensor frame w.r.t. a global frame (G) that is fixed w.r.t. the environment.

The accelerometer measurements at time $t_{k}$, where the integer *k* is used as a sample index, can be modeled as
(1)$$\begin{array}{cc} y_{a}^{S}\left(t_{k}\right) & =R^{SG}\left(t_{k}\right)\left(a^{G}\left(t_{k}\right)+g^{G}\right)+b_{a}^{S}+e_{a}^{S}\left(t_{k}\right), \end{array}$$
where $a^{G}\in R^{3}$ is the acceleration of the sensor w.r.t. the global frame and $g^{G}\in R^{3}$ is the gravitational acceleration, which is assumed to be constant in the environment. The measurements are corrupted by a constant additive bias $b_{a}^{S}$ and noise $e_{a}^{S}\left(t_{k}\right)\in R^{3}$, which is assumed to be Gaussian $e_{a}^{S}\left(t_{k}\right)∼N\left(0,\Sigma_{a}\right)$, with zero mean and covariance matrix $\Sigma_{a}$. The superscript *S* and *G* are used to denote in which reference frame a quantity is expressed in, and the rotation matrix $R^{SG}$ describes the rotation from the global frame to the sensor frame, i.e., we have that
(2)$$\begin{array}{cc} R^{SG}\left(t_{k}\right)\left(a^{G}\left(t_{k}\right)+g^{G}\right) & =a^{S}\left(t_{k}\right)+g^{S}\left(t_{k}\right). \end{array}$$

The multiplication between a rotation matrix and a vector is equivalent to a change of orthonormal basis.

The gyroscope measurements are modeled as
(3)$$\begin{array}{cc} y_{\omega }^{S}\left(t_{k}\right) & =R^{SG}\left(t_{k}\right)\omega^{G}\left(t_{k}\right)+b_{\omega }^{S}+e_{\omega }^{S}\left(t_{k}\right), \end{array}$$
where $\omega^{G}\in R^{3}$ is the angular velocity of the sensor frame in the global frame. Similar to the accelerometer, the measurements are corrupted by constant additive bias $b_{\omega }^{S}$ and noise $e_{\omega }^{S}\left(t_{k}\right)\in R^{3}$, which is assumed to be zero-mean Gaussian $e_{\omega }^{S}\left(t_{k}\right)∼N\left(0,\Sigma_{\omega }\right)$. Note that the same rotation matrix $R^{SG}$ as in (Equation 1) is used to rotate quantities from the global frame to the sensor frame because the accelerometer and the gyroscope are contained in the same IMU and their axes are assumed to be aligned. The gyroscope bias term $b_{\omega }^{S}$ can be compensated for through pre-calibration of the gyroscopes [24]. In Section 7.5, we will evaluate the effect of uncompensated biases on the proposed method.

Biases and Gaussian measurement noise have been shown to be the dominating error sources, even for low-cost IMUs [25]. However, for longer experiments or for low-quality IMUs, there are other types of errors that may need to be considered. These errors can still be well compensated for by pre-calibration or by online auto-calibration methods. Therefore, we only consider biases in our models, as these are the dominating systematic errors. The bias terms $b_{a}$ and $b_{\omega }$ are not constant, but drift slowly over time [26]. Sensor manufacturers typically provide a bias stability metric for their sensors, which tells the user the expected rate of the bias drift. Bias instability in inertial sensors is primarily caused by low-frequency flicker noise in the electronics and temperature fluctuations [27]. If the bias drift is significant enough that it needs to be compensated for, there are methods that model the biases as time or temperature dependent, enabling continuous estimation of drifting biases (see, e.g., [28,29]). Such methods can be used in combination with the method proposed in this paper. Low-quality IMUs may be affected by other systematic errors such as non-unit scale factors and misalignments/non-orthogonalities in the sensor axes. If the effect from these types of errors are non-negligible, it is advised to perform a more sophisticated pre-calibration of the sensors to compensate for these errors. Methods for in-field pre-calibration of such errors exist; see, e.g., [30,31,32,33].

## 3. Kinematics

The kinematic model of the hinge joint system has been described in previous works [12,15,16], and is recapitulated here in Section 3.1 and Section 3.2 for completeness.

### 3.1. Kinematic Constraints of Two Segments in a Kinematic Chain

Consider the kinematic chain model where we have two rigid body segments connected by a joint. The joint can have 1, 2 or 3 degrees of freedom (DOF). Furthermore, consider the case where each segment has one IMU rigidly attached to it in an arbitrary position and orientation. We therefore have two sensor frames, denoted by $S_{1}$ and $S_{2}$, that are fixed in the center of the accelerometer triad of each IMU. The DOF of the joint determines how many angles that are required to describe the orientation of $S_{2}$ w.r.t. $S_{1}$ and vice versa. We let subscripts i∈{1,2} denote quantities belonging to a specific sensor frame. Rigid body kinematics gives
(4)$$\begin{array}{cc} a_{i}^{S_{i}}\left(t\right) & =a_{0}^{S_{i}}\left(t\right)+\omega_{i}^{S_{i}}\left(t\right)\times \left(\omega_{i}^{S_{i}}\left(t\right)\times r_{i}^{S_{i}}\right)+{\dot{\omega }}_{i}^{S_{i}}\left(t\right)\times r_{i}^{S_{i}}, \end{array}$$
where $a_{i}$ are the accelerations of the sensor frames with i∈{1,2}, $a_{0}$ is the acceleration of the joint center, $\omega_{i}$ and ${\dot{\omega }}_{i}$ are the angular velocities and angular accelerations of the sensor frames and *t* is used to denote time-dependence of the kinematic variables. The positions of the joint center with respect to each sensor frame are denoted by $r_{i}$, which we assume to be unknown and constant for each sensor. All quantities in (Equation 4) are vectors in $R^{3}$ since they describe 3D motion. The acceleration of the joint center expressed in either of the sensor frames has the same magnitude but a different orientation. We have that
(5)$$\begin{array}{cc} a_{0}^{G}\left(t\right) & =R^{GS_{1}}\left(t\right)a_{0}^{S_{1}}\left(t\right)=R^{GS_{2}}\left(t\right)a_{0}^{S_{2}}\left(t\right) \end{array}$$
where $R^{GS_{i}}$ are the rotation matrices that maps a vector expressed in $S_{i}$ into the global frame.

For convenience we shall for the remainder of this document drop the use of the superscripts except for where it’s needed. Hence, the sensor frame of a kinematic variable will be given by subscript i∈{1,2}. We will also drop the use of *t* to denote time-dependence unless we want to refer to the kinematic variables at specific time instances. The relationship in (Equation 4) is linear in $a_{0}$ and $r_{i}$ and can equivalently be formulated as
(6)$$\begin{array}{cc} a_{i} & =a_{0}^{S_{i}}+K\left(\omega_{i},{\dot{\omega }}_{i}\right)r_{i}, \end{array}$$
where
(7)$$\begin{array}{c} K\left(\omega ,\dot{\omega }\right)=\left[\begin{array}{ccc} -\omega_{y}^{2}-\omega_{z}^{2} & \omega_{x}\omega_{y}-{\dot{\omega }}_{z} & \omega_{x}\omega_{z}+{\dot{\omega }}_{y}\\ \omega_{x}\omega_{y}+{\dot{\omega }}_{z} & -\omega_{x}^{2}-\omega_{z}^{2} & \omega_{y}\omega_{z}-{\dot{\omega }}_{x}\\ \omega_{x}\omega_{z}-{\dot{\omega }}_{y} & \omega_{y}\omega_{z}+{\dot{\omega }}_{x} & -\omega_{x}^{2}-\omega_{y}^{2} \end{array}\right], \end{array}$$
and where subscripts x,y,z denote the elements of the three-dimensional vectors. For convenience of notation we will write $K_{i}=K\left(\omega_{i},{\dot{\omega }}_{i}\right)$.

### 3.2. Kinematic Constraints of a Hinge Joint System

For a 1-DOF joint, the two segments can only rotate independently with respect to each other along the joint axis. We let ∥·∥ denote the Euclidean vector norm, then the joint axis is defined by the unit vector $j\in R^{3},∥j∥=1$. We refer to such a joint as a hinge joint. We let $j_{1}$ and $j_{2}$ denote the direction of the joint axis in the respective sensor frames. Since the two IMUs are assumed to be rigidly attached to the segments, $j_{1}$ and $j_{2}$ are constant. The joint axis *j* expressed in the global frame must then satisfy
(8)$$\begin{array}{cc} j^{G}\left(t\right) & =R^{GS_{1}}\left(t\right)j_{1}^{S_{1}}=R^{GS_{2}}\left(t\right)j_{2}^{S_{2}}, \end{array}$$
meaning that the vectors $j_{i}$ expressed in the two sensor frame has the same direction as *j* in the global frame, see Figure 1, and time-dependence is only caused by the rotations of the sensor frames in the global frame. We can decompose the angular velocities into one component that is parallel to the joint axis and one that is perpendicular to the joint axis
(9)$$\begin{array}{cc} \omega_{i} & =\omega_{j_{i}}+\omega_{j_{i}^{⊥}}, \end{array}$$
(10)$$\begin{array}{cc} \omega_{j_{i}} & =j_{i}^{⊤}\omega_{i}j_{i}, \end{array}$$
(11)$$\begin{array}{cc} \omega_{j_{i}^{⊥}} & =\omega_{i}-\omega_{j_{i}}=\omega_{i}-j_{i}^{⊤}\omega_{i}j_{i}. \end{array}$$

Since the two segments can only rotate independently along the joint axis, it follows that the perpendicular components must have the same magnitude regardless of reference frame
(12)$$\begin{array}{cc} ∥\omega_{j_{1}^{⊥}}∥ & =∥\omega_{j_{2}^{⊥}}∥. \end{array}$$

The magnitude of the perpendicular component can also be computed from the cross product between the angular velocity and the joint axis(13)$$\begin{array}{cc} ∥\omega_{i}-j_{i}^{⊤}\omega_{i}j_{i}∥ & =∥\omega_{i}\times j_{i}∥. \end{array}$$

Combining (Equation 12) and (Equation 13) we formulate the angular velocity constraint
(14)$$\begin{array}{c} ∥\omega_{1}\times j_{1}∥-∥\omega_{2}\times j_{2}∥=0, \end{array}$$
which must be satisfied by hinge joint systems.

Looking at the projection of the accelerations onto the joint axis, from (Equation 6) we have that
(15)$$\begin{array}{cc} j_{i}^{⊤}a_{i}j_{i} & =j_{i}^{⊤}a_{0}^{S_{i}}j_{i}+j_{i}^{⊤}K_{i}r_{i}j_{i}. \end{array}$$

Because $j_{i}$ has the same direction as *j* in the global frame, it must also be the same for the projection of $a_{0}$ onto $j_{i}$, it follows from (Equation 5) and (Equation 8) that
(16)$$\begin{array}{cc} {j^{G}}^{⊤}a_{0}^{G}j^{G} & =R^{GS_{1}}j_{1}j_{1}^{⊤}a_{0}^{S_{1}}=R^{GS_{2}}j_{2}j_{2}^{⊤}a_{0}^{S_{2}}\\ \; & \Rightarrow j_{1}^{⊤}a_{0}^{S_{1}}=j_{2}^{⊤}a_{0}^{S_{2}}. \end{array}$$

By projecting the accelerations onto the joint axis and subtracting one from the other we get
(17)$$\begin{array}{c} \begin{array}{cc} j_{1}^{⊤}a_{1}-j_{2}^{⊤}a_{2} & =j_{1}^{⊤}a_{0}^{S_{1}}-j_{2}^{⊤}a_{0}^{S_{2}}+j_{1}^{⊤}K_{1}r_{1}-j_{2}^{⊤}K_{2}r_{2}\\ \; & =j_{1}^{⊤}K_{1}r_{1}-j_{2}^{⊤}K_{2}r_{2}, \end{array} \end{array}$$
where we see that only the rotational components of the accelerations remain on the right hand side. The relationship (Equation 17) is the exact acceleration constraint of the hinge joint system. The right hand side (r.h.s.) of (Equation 17) is zero if and only if either $K_{i}r_{i}⊥j_{i}$ or $K_{i}=0$ are satified for all i∈{1,2}. It is clear that if the rotational acceleration components along the direction of the joint axis are small ($j_{i}^{⊤}K_{i}r_{i}\approx 0,\;\forall i$), the r.h.s. will vanish
(18)$$\begin{array}{cc} j_{1}^{⊤}a_{1}-j_{2}^{⊤}a_{2} & \approx 0, \end{array}$$
which forms the approximate acceleration constraint for the hinge joint system.

## 4. Joint Axis Estimation

We assume that we have two IMUs, one attached to each segment of a hinge joint system. Measurements from a completely unspecified motion has been collected. We will use $y_{\omega ,i}$ to refer to the gyroscope measurements (Equation 3) and $y_{a,i}$ to refer to the accelerometer measurements (Equation 1) from Sensor i∈{1,2}. We will use the non-indexed $y_{\omega }$ and $y_{a}$ to refer to measurements from both sensors as
(19)$$\begin{array}{cc} y_{\omega } & ={\left[\begin{array}{cc} y_{\omega ,1}^{⊤} & y_{\omega ,2}^{⊤} \end{array}\right]}^{⊤}, \end{array}$$
and similarly for $y_{a}$. We let $D^{N}=\left\{y_{\omega }^{N},y_{a}^{N}\right\}$ denote our data, which consists of *N* samples of recorded motion. Each sample in the data set is assigned a sample index k∈{1,…,N}, such that $t_{k}$ refers to the sampling time of the *k*th measurement relative to the beginning of the recorded motion.

Given the data $D^{N}$ from the two IMUs, the variables we want to estimate are the unit vectors $j_{i}$ which corresponds to the directions of the joint axis *j* in the two sensor frames. We let ${\hat{j}}_{i}$ denote the estimate of $j_{i}$. Note that the joint axis in one sensor frame can be described by either $\pm j_{i}$ since a clockwise rotation w.r.t. the positive axis is equivalent to a counter-clockwise rotation w.r.t. the negative axis. However, we require both $j_{1}$ and $j_{2}$ to have the same sign (direction) to correspond to either ±j in the global frame, otherwise a clockwise rotation for one sensor might be considered a counter-clockwise rotation for the other sensor and vice versa. That is, the sign pairing of the joint axes in the sensor coordinate frames is important. Consequently, $\left(\pm j_{1},\pm j_{2}\right)$ is the correct sign pairing and $\left(\pm j_{1},\mp j_{2}\right)$ is the wrong sign pairing.

### 4.1. Formulating the Optimization Problem

We parametrize $j_{i}$ using spherical coordinates to enforce the unit vector constraint
(20)$$\begin{array}{cc} x & ={\left[\begin{array}{cccc} \theta_{1} & \phi_{1} & \theta_{2} & \phi_{2} \end{array}\right]}^{⊤}, \end{array}$$
(21)$$\begin{array}{cc} j_{i}\left(x\right) & =\left[\begin{array}{c} \cos\theta_{i}\cos\phi_{i}\\ \cos\theta_{i}\sin\phi_{i}\\ \sin\theta_{i} \end{array}\right], \end{array}$$
which then become the unknown parameters to estimate. The estimation problem for the joint axis is formulated as
(22)$$\begin{array}{cc} \hat{x} & =\underset{x}{arg min}V\left(x\right), \end{array}$$
(23)$$\begin{array}{cc} V(x) & =\sum_{k=1}^{N}{\left[e_{\omega }\left(k,x\right)\right]}^{2}+{\left[e_{a}\left(k,x\right)\right]}^{2}, \end{array}$$
where $e_{\omega }\left(k,x\right)$ and $e_{a}\left(k,x\right)$ are scalar residual terms, based on the angular velocity constraint (Equation 14) and acceleration constraints (Equation 18) of the hinge joint system
(24)$$\begin{array}{cc} e_{\omega }\left(k,x\right) & =w_{\omega }[∥y_{\omega ,1}\left(t_{k}\right)\times j_{1}\left(x\right)∥-∥y_{\omega ,2}\left(t_{k}\right)\times j_{2}\left(x\right)∥], \end{array}$$
(25)$$\begin{array}{cc} e_{a}\left(k,x\right) & =w_{a}\left[j_{1}^{⊤}\left(x\right)y_{a,1}\left(t_{k}\right)-j_{2}^{⊤}\left(x\right)y_{a,1}\left(t_{k}\right)\right]. \end{array}$$

Two scalars $w_{\omega }$ and $w_{a}$ are used to change the relative weighting of the residuals.

### 4.2. Identifiability and Local Minima

For the gyroscope measurements to contain information about the joint axis, they have to be recorded from motions where the joint angle is excited, i.e., when the two segments rotate independently. These motions should contain either simultaneous planar rotations, where the segments rotate simultaneously in the plane perpendicular to the joint axis, or sequential rotations of the segments. However, stiff joint motions, which can have a significant angular rate but no independent rotation of the segments, do not facilitate identifiability of the joint axis [16]. For the non-informative stiff joint motions, the relative rotation of the two sensors can be described by a time-invariant rotation matrix *R* and we have that
(26)$$\begin{array}{cc} ∥\omega_{2}\left(t_{k}\right)\times j_{2}∥ & =∥R\left(\omega_{1}\left(t_{k}\right)\times j_{1}\right)∥=∥\omega_{1}\left(t_{k}\right)\times j_{1}∥, \end{array}$$
where we see that for any choice of $j_{1}$, the vector $j_{2}=Rj_{1}$ will minimize the gyroscope residual (Equation 24). Therefore, we want motions where $∥\omega_{1}\left(t_{k}\right)∥\neq ∥\omega_{2}\left(t_{k}\right)∥$, which implies that the segments are rotating independently and we require motions where $∥\omega_{i}\left(t_{k}\right)∥>0$ for at least some time, since $∥\omega_{i}\left(t_{k}\right)∥=0\Rightarrow ∥\omega_{i}\left(t_{k}\right)\times j_{i}∥=0,\forall j_{i}$.

If only acceleration information is considered, we get the following over-determined system of linear equations
(27)$$\begin{array}{cc} \underset{=A}{\underset{︸}{\left[\begin{array}{cc} a_{1}^{⊤}\left(t_{1}\right) & -a_{2}^{⊤}\left(t_{1}\right)\\ \vdots & \vdots \\ a_{1}^{⊤}\left(t_{M}\right) & -a_{2}^{⊤}\left(t_{M}\right) \end{array}\right]}}\left[\begin{array}{c} j_{1}\\ j_{2} \end{array}\right] & =0, \end{array}$$
which has a unique solution if rank(A)=5, in which case ${\left[\begin{array}{cc} j_{1}^{⊤} & j_{2}^{⊤} \end{array}\right]}^{⊤}$ lies in the null-space of *A*. This holds when the acceleration constraint holds exactly for all $t_{k}$, the accelerations measured are exact and the angular rate and angular accelerations of the sensors are parallel with *j* [16]. Therefore, for the accelerometer, we want measurements that increase the separation between the column-space and the null-space of *A*.

The proposed method uses both gyroscope and accelerometer information, and their relative contribution to the cost function is controlled by the weight parameters $w_{\omega }$ and $w_{a}$. Figure 2 shows how the weights affect the cost function in the case that $w_{a}=1$ and $w_{\omega }$ is allowed to vary. For small $w_{\omega }$, the local minima corresponds to the correct sign pairing $\left(\pm j_{1},\pm j_{2}\right)$, whereas the local maxima corresponds to the wrong sign pairing $\left(\pm j_{1},\mp j_{2}\right)$. Note that each local minimum is equally valid for small $w_{\omega }$ because of the periodicity of the spherical coordinates. The acceleration residuals are relatively large whereas the gyroscope residuals are relatively small at the locations corresponding to the wrong sign pairing. Therefore, as $w_{\omega }$ increases the gyroscope residuals will contribute more to the cost function. The peaks associated with the wrong sign pairing are flattened and new local minima will eventually appear at these locations. Therefore, for large $w_{\omega }$ an optimization method (solver) can end up in the wrong local minimum. However, regardless of which sign pairing the solver finds, the opposite sign pairing can always be obtained at $x={\left[\begin{array}{cccc} \theta_{1} & \phi_{1} & -\theta_{2} & \phi_{2}+\pi \end{array}\right]}^{⊤}$. Therefore, if our solver finds the estimate ${\hat{x}}^{\left(1\right)}$ we can reinitialize at
(28)$$\begin{array}{c} {\left[\begin{array}{cccc} {\hat{\theta }}_{1}^{\left(1\right)} & {\hat{\phi }}_{1}^{\left(1\right)} & -{\hat{\theta }}_{2}^{\left(1\right)} & {\hat{\phi }}_{2}^{\left(1\right)}+\pi \end{array}\right]}^{⊤}, \end{array}$$
and obtain a new estimate ${\hat{x}}^{\left(2\right)}$. Then we select the local minimum with the smallest value of the cost function as our estimate
(29)$$\begin{array}{cc} \hat{x} & =\underset{x\in \{{\hat{x}}^{\left(1\right)},{\hat{x}}^{\left(2\right)}\}}{arg min}V\left(x\right). \end{array}$$

Therefore, it is possible to find the correct sign pairing as long as $V({\hat{x}}^{\left(2\right)})$ is numerically distinguishable from $V({\hat{x}}^{\left(1\right)})$. As discussed in this section and shown in Figure 2, the relative weighting of the residuals determines how easy it is to distinguish a correct local minimum from a wrong one. If $w_{\omega }$ is set to be significantly larger than $w_{a}$, we expect the acceleration residuals to eventually become so small relative to the gyroscope residuals, that the solver is no longer sensitive enough to detect the difference between correct and wrong local minima.

### 4.3. Solving the Optimization Problem

The optimization problem (Equation 22) is a nonlinear least-squares problem. An efficient solver for such problems is the Gauss–Newton method [34]. Given an initial estimate $\hat{x}\left(0\right)$ the Gauss–Newton method iteratively updates the estimate according to    
(30)$$\begin{array}{c} \begin{array}{cc} \hat{x}\left(k+1\right) & =\hat{x}\left(k\right)-\alpha {\left(J^{⊤}\left(\hat{x}\left(k\right)\right)J\left(\hat{x}\left(k\right)\right)\right)}^{-1}J^{⊤}\left(\hat{x}\left(k\right)\right)e\left(\hat{x}\left(k\right)\right)\\ \; & =\hat{x}\left(k\right)-\alpha \Delta_{x}\left(k\right) \end{array}, \end{array}$$
where *k* is only used here as an integer index denoting the iterations of the method and is not to be confused with the sample index. The method uses the Jacobian matrix $J\left(x\right)\in R^{2N\times 4}$, which contains all first-order partial derivatives of $e_{\omega }$ and $e_{a}$
(31)$$\begin{array}{cc} J(x) & =\left[\begin{array}{cccc} \frac{\partial e_{\omega }\left(1,x\right)}{\partial \theta_{1}} & \frac{\partial e_{\omega }\left(1,x\right)}{\partial \phi_{1}} & \frac{\partial e_{\omega }\left(1,x\right)}{\partial \theta_{2}} & \frac{\partial e_{\omega }\left(1,x\right)}{\partial \phi_{1}}\\ \vdots & \vdots & \vdots & \vdots \\ \frac{\partial e_{\omega }\left(N,x\right)}{\partial \theta_{1}} & \frac{\partial e_{\omega }\left(N,x\right)}{\partial \phi_{1}} & \frac{\partial e_{\omega }\left(N,x\right)}{\partial \theta_{2}} & \frac{\partial e_{\omega }\left(N,x\right)}{\partial \phi_{1}}\\ \frac{\partial e_{a}\left(1,x\right)}{\partial \theta_{1}} & \frac{\partial e_{a}\left(1,x\right)}{\partial \phi_{1}} & \frac{\partial e_{a}\left(1,x\right)}{\partial \theta_{2}} & \frac{\partial e_{a}\left(1,x\right)}{\partial \phi_{1}}\\ \vdots & \vdots & \vdots & \vdots \\ \frac{\partial e_{a}\left(N,x\right)}{\partial \theta_{1}} & \frac{\partial e_{a}\left(N,x\right)}{\partial \phi_{1}} & \frac{\partial e_{a}\left(N,x\right)}{\partial \theta_{2}} & \frac{\partial e_{a}\left(N,x\right)}{\partial \phi_{1}} \end{array}\right], \end{array}$$
and $e\left(x\right)\in R^{2N}$ is the residual vector
(32)$$\begin{array}{cc} e(x) & ={\left[\begin{array}{cccccc} e_{\omega }\left(1,x\right) & \ldots & e_{\omega }\left(N,x\right) & e_{a}\left(1,x\right) & \ldots & e_{a}\left(N,x\right) \end{array}\right]}^{⊤}. \end{array}$$

The term ${\left(J^{⊤}\left(x\right)J\left(x\right)\right)}^{-1}$ is an approximation of the Hessian of V(x), which is given by
(33)$$\begin{array}{c} \frac{d^{2}V\left(x\right)}{dx^{2}}=J^{⊤}\left(x\right)J\left(x\right)+\sum_{k=1}^{N}e_{\omega }\left(k,x\right)\frac{d^{2}e_{\omega }\left(k,x\right)}{dx^{2}}\\ \;+\sum_{k=1}^{N}e_{a}\left(k,x\right)\frac{d^{2}e_{a}\left(k,x\right)}{dx^{2}} \end{array},$$
where the higher-order terms are ignored, yielding
(34)$$\begin{array}{c} \frac{d^{2}V\left(x\right)}{dx^{2}}\approx J^{⊤}\left(x\right)J\left(x\right). \end{array}$$

The partial derivatives of the residuals (Equation 24) and (Equation 25) in the Jacobian (Equation 31) are computed in the following way using the chain rule
(35)$$\begin{array}{cc} \frac{\partial e_{\omega }\left(k,x\right)}{\partial x} & =\frac{\partial j}{\partial x}\frac{\partial e_{\omega }\left(k,x\right)}{\partial j}w_{\omega }\left(k\right), \end{array}$$
(36)$$\begin{array}{cc} \frac{\partial e_{\omega }\left(k,x\right)}{\partial j} & =\left[\begin{array}{c} \frac{\partial (∥y_{\omega ,1}\left(t_{k}\right)\times j_{1}\left(x\right)∥)}{\partial j_{1}}\\ -\frac{\partial (∥y_{\omega ,2}\left(t_{k}\right)\times j_{2}\left(x\right)∥)}{\partial j_{2}} \end{array}\right], \end{array}$$
(37)$$\begin{array}{cc} \frac{\partial (∥y_{\omega ,i}\left(t_{k}\right)\times j_{i}\left(x\right)∥)}{\partial j_{i}} & =\frac{\left(y_{\omega ,i}\left(t_{k}\right)\times j_{i}\right)\times y_{\omega ,i}\left(t_{k}\right)}{∥y_{\omega ,i}\left(t_{k}\right)\times j_{i}\left(x\right)∥}, \end{array}$$
(38)$$\begin{array}{cc} \frac{\partial e_{a}\left(k,x\right)}{\partial x} & =\frac{\partial j}{\partial x}\frac{\partial e_{a}\left(k,x\right)}{\partial j}, \end{array}$$
(39)$$\begin{array}{cc} \frac{\partial e_{a}\left(k,x\right)}{\partial j} & =\left[\begin{array}{c} y_{a,1}\left(t_{k}\right)\\ -y_{a,2}\left(t_{k}\right) \end{array}\right]w_{a}\left(k\right), \end{array}$$
(40)$$\begin{array}{cc} \frac{\partial j}{\partial x} & =\left[\begin{array}{cc} \frac{\partial j_{1}}{\partial x_{1}} & 0\\ 0 & \frac{\partial j_{2}}{\partial x_{2}} \end{array}\right], \end{array}$$
(41)$$\begin{array}{cc} \frac{\partial j_{i}}{\partial x_{i}} & ={\left[\begin{array}{cc} -\sin\theta_{i}\cos\phi_{i} & -\cos\theta_{i}\sin\phi_{i}\\ -\sin\theta_{i}\sin\phi_{i} & \cos\theta_{i}\cos\phi_{i}\\ \cos\theta_{i} & 0 \end{array}\right]}^{⊤}. \end{array}$$

The term $\Delta_{x}$ in (Equation 30) defines the search direction, and $-\Delta_{x}$ is a descent direction, meaning that moving our estimate in that direction will decrease the value of the cost function. The scalar 0<α≤1 is known as the step length, which controls how far our estimates move in the descent direction. By using a method known as backtracking line search [35], we find a value for α that is guaranteed to lower the value of the cost function. If no such α is found or the change in the value of V(x) is too small, below a set tolerance level $V_{\mathrm{tol}}$, the Gauss–Newton method terminates and returns the estimate corresponding to the current iteration $\hat{x}=\hat{x}\left(k\right)$.

The complete joint axis estimation method, including the steps of the Gauss–Newton method and the re-initialization step (Equation 28) required to identify the minimum corresponding to the correct sign pairing, is described in Algorithm 1.
**Algorithm 1** Joint axis estimation**Require:** Data $D^{N}=\left\{y_{\omega }^{N},y_{a}^{N}\right\}$, initial estimate $\hat{x}\left(0\right)$, tolerance $V_{\mathrm{tol}}$, residual weights $w_{\omega }$ and $w_{a}$.
1:**for** i∈{1,2} **do**2:    k←0.                      ▹ Begin Gauss–Newton.3:    $\Delta_{V}\leftarrow V_{\mathrm{tol}}$.4:    $V\left(0\right)\leftarrow V(\hat{x}\left(0\right))$.                 ▹V(x) defined by (Equation 23).5:    **while**
$\Delta_{V}\geq V_{\mathrm{tol}}$
**do**6:        Compute the Jacobian $J(\hat{x}\left(k\right))$ and the residuals $e(\hat{x}\left(k\right))$ according to (Equation 31) and (Equation 32).7:        $\Delta_{x}\left(k\right)\leftarrow {\left(J^{⊤}\left(\hat{x}\left(k\right)\right)J\left(\hat{x}\left(k\right)\right)\right)}^{-1}J^{⊤}\left(\hat{x}\left(k\right)\right)e\left(\hat{x}\left(k\right)\right)$.8:        Obtain step length α using backtracking line search.9:        $\hat{x}\left(k+1\right)\leftarrow \hat{x}\left(k\right)-\alpha \Delta_{x}\left(k\right)$.10:        k←k+1.11:        $V\left(k\right)\leftarrow V(\hat{x}\left(k\right))$.12:        $\Delta_{V}\leftarrow \left|V\left(k-1\right)-V\left(k\right)\right|$.13:    **end while**14:    $\hat{x}\leftarrow \hat{x}\left(k\right)$.                     ▹ End Gauss–Newton.15:    ${\hat{x}}^{\left(i\right)}={\left[\begin{array}{cccc} {\hat{\theta }}_{1}^{\left(i\right)} & {\hat{\phi }}_{1}^{\left(i\right)} & {\hat{\theta }}_{2}^{\left(i\right)} & {\hat{\phi }}_{2}^{\left(i\right)} \end{array}\right]}^{⊤}\leftarrow \hat{x}$.16:    $\hat{x}\left(0\right)\leftarrow {\left[\begin{array}{cccc} {\hat{\theta }}_{1}^{\left(i\right)} & {\hat{\phi }}_{1}^{\left(i\right)} & -{\hat{\theta }}_{2}^{\left(i\right)} & {\hat{\phi }}_{2}^{\left(i\right)}+\pi \end{array}\right]}^{⊤}$.        ▹ Initialize at $-{\hat{j}}_{2}$.17:**end for**18:$\hat{x}\leftarrow {arg min}_{x\in \{{\hat{x}}^{\left(1\right)},{\hat{x}}^{\left(2\right)}\}}V\left(x\right)$.               ▹Correct sign pairing.19:**return** $j(\hat{x})$.


## 5. Sample Selection

A key feature of plug-and-play estimation is that it should not require specific calibration data, recorded from predetermined motions. Rather, such plug-and-play methods should be able to use data recorded from arbitrary motions. Such data sets could be very large, and using all available data for identification is often unnecessary and resource-demanding. It is also possible that very few samples in the data set contain information about the joint axis. In a sense, too much bad information might ruin the good information. To handle this, we propose a method for selecting samples to use for estimation.

In the following sections we assume that we want a maximum of $N_{\max}$ gyroscope and accelerometer measurements can be used to identify the joint axis, but that we have $N>N_{\max}$ measurements available to us to choose from.

### 5.1. Gyroscope

To distinguish between informative and non-informative motions, we use the difference in angular velocity magnitude measured by the two gyroscopes    
(42)$$\begin{array}{cc} \Delta_{\omega }\left(k\right) & =∥y_{\omega ,1}\left(t_{k}\right)∥-∥y_{\omega ,2}\left(t_{k}\right)∥, \end{array}$$
which is a sufficient metric for detecting independent rotations of the sensors, and hence the two segments. For stationary segments $\Delta_{\omega }\left(k\right)=0$. One thing to note is that $\Delta_{\omega }\left(k\right)$ cannot differentiate between informative motions where $∥\omega_{1}∥\approx ∥\omega_{2}∥$ and non-informative stiff joint rotations. For example, the two segments can undergo simultaneous planar rotations, where the two segments rotate in different directions but with approximately the same magnitude. However, for realistic motions, especially for motions performed by humans, it is unlikely that independent rotations will have the same magnitude, even for short moments.

Each gyroscope measurement is given a score
(43)$$\begin{array}{cc} s_{\omega }\left(k\right) & =\Delta_{\omega }\left(l^{\prime }\right) \end{array}$$
(44)$$\begin{array}{cc} l^{\prime } & =\underset{l}{arg min}\left|\Delta_{\omega }\left(l\right)\right|,\;l\in \left(k-n,k+n\right) \end{array}$$
that is equal to the $\Delta_{\omega }$ with smallest magnitude in a window of 2n+1 samples. This is to avoid selecting large outliers of $\Delta_{\omega }$. For example, if the system is not completely rigid or the sensors are not rigidly attached, the kinematic constraints are violated, and some samples of stiff joint motion can obtain a large $\Delta_{\omega }$ value. However, if the outliers are relatively few, there should be $\Delta_{\omega }$ with smaller magnitude among neighboring samples. In some sense, $s_{\omega }\left(k\right)$ assumes a conservative score for each sample.

When the score $s_{\omega }$ has been computed for all measurements, the list of measurements is sorted in descending order such that $s_{\omega }\left(k^{\prime }\right)\geq s_{\omega }\left(k^{\prime }+1\right),\forall k^{\prime }\in \left(1,N-1\right)$, where $k^{\prime }$ is a new index variable used to denote the sorted order. The first and last $N_{\max}/2$ of the sorted gyroscope measurements are selected, or, equivalently, the measurements corresponding to the middle of the list, i.e., with index $k^{\prime }\in \left(N_{\max}/2+1,N-N_{\max}/2\right)$ are removed from the set of measurements. By doing this, the algorithm will make sure that measurements with excitation in both sensors are selected, since $\Delta_{\omega }>0$ means that Sensor 1 has larger angular rate than Sensor 2 and vice versa for $\Delta_{\omega }<0$. The gyroscope sample selection method is described in Algorithm 2. In essence, the algorithm picks half the required points from either end of the sorted list.
**Algorithm 2** Gyroscope sample selection**Require:** Gyroscope data $y_{\omega }^{N}$, number of allowed measurements, $N_{\max}$, window size *n*.
1:**if** $N>N_{\max}$ **then**2:    Compute $s_{\omega }\left(k\right),\forall k$ according to (Equation 43).3:    Obtain the sorted order $k^{\prime }$ such that $s_{\omega }\left(k^{\prime }\right)\leq s_{\omega }\left(k^{\prime }+1\right),\;\forall k^{\prime }\in \left(1,N-1\right)$.4:    Remove the $N-N_{\max}$ samples $y_{\omega }\left(t_{k}\right),\forall k^{\prime }\in \left(N_{\max}/2+1,N-N_{\max}/2\right)$ from $y_{\omega }$.5:**end if**6:**return** $y_{\omega }^{N_{\max}}$


### 5.2. Accelerometer

The acceleration constraint is accurate when the angular rate and angular accelerations are small, since that makes the right hand side of (Equation 17) vanish. Note that linear acceleration terms in (Equation 17), which are collected in $a_{0}$, always cancel out. Therefore, we do not use the energy of the accelerometer measurements to determine if the acceleration constraint is valid. Instead, we give each acceleration measurement a penalty based on the average angular rate energy
(45)$$\begin{array}{cc} E_{i}\left(k\right) & =\left\{\begin{array}{cc} \frac{1}{2n+1}\sum_{l=k-n}^{k+n}{∥y_{\omega ,i}\left(t_{l}\right)∥}^{2}, & \;n<k\leq N-n\\ \infty , & \;\mathrm{otherwise} \end{array}\right., \end{array}$$
where the average is calculated from a window of size 2n+1, centered around each sample. This angular rate energy statistic has been shown to be an effective detector of stationarity in foot-mounted inertial navigation [36], so-called zero-velocity detection.

Small $E_{i}\left(k\right)$ indicate that Sensor *i* is stationary. For the hinge joint system, it is sufficient for one sensor to be stationary since the acceleration components in the plane normal to the joint axis does not change the r.h.s of (Equation 17). If one sensor is stationary, then the other sensor can only have accelerations that are induced by independent rotation, which has to be in the plane. For this reason, the penalty given to each pair of acceleration measurements is chosen as
(46)$$\begin{array}{cc} s_{a}\left(k\right) & =\min\{E_{1}\left(t_{k}\right),E_{2}\left(t_{k}\right)\}. \end{array}$$

As a first step of the accelerometer sample selection, measurements with $s_{a}\left(k\right)>E_{\mathrm{th}}$ are removed, where $E_{\mathrm{th}}$ is a scalar threshold parameter, which should be chosen to remove measurements for which it is likely that the motion violates the acceleration constraint.

We also need to consider the conditions for identifiability of the joint axis. That is, we want our measurements to increase the separation between the column-space and the null-space of the matrix *A* in (Equation 27). In practice, *A* will have full rank regardless of the motion, since the measurements are corrupted by noise and bias and the acceleration constraint does not hold for arbitrary motions. However, if *A* has one singular value that is relatively small compared to the other singular values, it can be considered to be approximately rank 5. Consider the singular value decomposition (SVD) of *A*
(47)$$\begin{array}{c} A=U\Sigma W^{⊤}, \end{array}$$
where the diagonal elements $\sigma_{1}$ to $\sigma_{6}$ of $\Sigma \in R^{M\times 6}$ are the singular values and the columns of *U* and *W* represents orthonormal bases in $R^{N}$ and $R^{6}$, respectively. The columns of *W* are known as the right-singular vectors of *A*, and each is associated with a corresponding singular value, i.e., if
(48)$$\begin{array}{cc} \mathrm{diag}(\Sigma ) & =\left[\begin{array}{cccccc} \sigma_{1} & \sigma_{2} & \sigma_{3} & \sigma_{4} & \sigma_{5} & \sigma_{6} \end{array}\right], \end{array}$$
(49)$$\begin{array}{cc} W & =\left[\begin{array}{cccccc} w_{1} & w_{2} & w_{3} & w_{4} & w_{5} & w_{6} \end{array}\right], \end{array}$$
the right-singular vector $w_{1}$ is associated with $\sigma_{1}$. The singular values are ordered $\sigma_{1}\geq \sigma_{2}\geq \ldots \geq \sigma_{6}\geq 0$. We have that $w_{1}$ is the direction in $R^{6}$ where the rows of *A* are most coherent, meaning that
(50)$$\begin{array}{cc} w_{1} & =\underset{w,∥w∥=1}{arg max}\left|Aw\right|, \end{array}$$
which has the interpretation that $w_{1}$ is the direction that is most separated from the null-space of *A*. The information about *j* that is contained in *A* is directly linked to the separation between the null-space and the column space of *A*. The intuition behind this can be seen by comparing the system of linear equations in (Equation 27) to the definition of $w_{1}$ in (Equation 50), where it appears most unlikely that *j* should be parallel with $w_{1}$. In fact, the least-squares estimator for *j* given by
(51)$$\begin{array}{cc} \hat{j} & =\underset{j}{arg min}{∥Aj∥}^{2}, \end{array}$$
has solutions on the line in $R^{6}$, which is spanned by $w_{6}$, the right-singular vector associated with the smallest singular value. If we add the constraints $∥j_{1}∥=∥j_{2}∥=1$ the two solutions with correct sign pairing, corresponding to $\left(j_{1},j_{2}\right)$ and $\left(-j_{1},-j_{2}\right)$ can be obtained through normalization. A problem arises when multiple singular values are close to zero, in which case the value of ${∥Aj∥}^{2}$ will be small in more than one direction, and the uncertainty in the estimate increases. If *A* is only allowed to have $N_{\max}$ rows, we should therefore only remove measurements whose rows in *A* are most coherent with $w_{1}$, the direction with most information. This way, we make sure that space is always allocated for measurements with rows that do not align with $w_{1}$, which over time should increase the discrepancy between the two smallest singular values and increase the certainty of the least-squares estimator.

The coherence between a row in *A* and the right-singular vector $w_{1}$ is computed as the vector $c\in R^{M}$, with the elements
(52)$$\begin{array}{cc} c_{k} & =\frac{|A_{k}w_{1}|}{∥A_{k}∥∥w_{1}∥}, \end{array}$$
(53)$$\begin{array}{cc} A_{k} & =\left[\begin{array}{cc} y_{a,1}^{⊤}\left(t_{k}\right) & -y_{a,2}^{⊤}\left(t_{k}\right) \end{array}\right], \end{array}$$
where $A_{k}$ is the $k^{\mathrm{th}}$ row vector in *A*, and $c_{k}$ has a value of 1 if $A_{k}$ is parallel to $w_{1}$ and 0 if they are orthogonal. A $c_{k}>0.5$ means that $A_{k}$ has most of its magnitude in the direction of $w_{1}$. Therefore, we choose to remove measurements with the largest $s_{a}\left(k\right)$ where $c_{k}>0.5$. This ensures that we also keep good measurements in the $w_{1}$ direction, while allocating space for measurements with new information about *j*. The algorithm for selecting accelerometer samples is described in Algorithm 3.
**Algorithm 3** Accelerometer sample selection**Require:** Data $D^{N}=\left\{y_{\omega }^{N},y_{a}^{N}\right\}$, number of allowed measurements $N_{\max}$, window size *n*, threshold $E_{\mathrm{th}}$.
1:**if** $N>N_{\max}$ **then**2:    Compute $s_{a}\left(k\right),\forall k$ according to (Equation 46) using window size *n*.3:    Remove measurements where $s_{a}\left(k\right)>E_{\mathrm{th}}$ from *a*.4:    $N\leftarrow |y_{a}|$.5:    **while**
$N>N_{\max}$
**do**6:        Compute the SVD A=UΣW, with *A* given by (Equation 27).7:        Compute the coherence *c* according to (Equation 52).8:        Remove the measurement with largest $s_{a}\left(k\right)$ where $c_{k}>0.5$ from *a*.9:        $N\leftarrow |y_{a}|$.      ▹ *A* changes in subsequent iterations.10:    **end while**11:**end if**12:**return** $y_{a}^{N_{\max}}$.


### 5.3. Online Implementation

The two proposed sample selection algorithms can be implemented for an online application. For Algorithm 2, simply save the scores $s_{\omega }$ and re-use them when a new batch of data is available, new $s_{\omega }$ only needs to be computed for the previously unseen measurements. The same principle holds for Algorithm 3 and $s_{a}$.

## 6. Uncertainty Quantification

When identifying an unknown quantity, it is useful for the user of the method to know if they can expect their estimate to be accurate given the data that is available, or if more informative data needs to be collected. Here we propose a method for quantifying both local and global uncertainty of an estimate $\hat{x}$.

The local uncertainty is obtained through estimating the covariance matrix of the estimation errors using the Jacobian of the cost function. Global uncertainty is obtained through solving multiple parallel or sequential optimization problems with different random initializations, then comparing the resulting estimates to see if they correspond to the same joint axis.

The local and global uncertainty metrics are combined into an algorithm that can be used to determine if a current estimate is of acceptable accuracy or if more informative data needs to be collected.

### 6.1. Local Uncertainty

We approximate the cost function V(x) (Equation 23) as a quadratic function near the estimate $\hat{x}$
(54)$$\begin{array}{c} \hat{V}\left(x\right)=V\left(\hat{x}\right)+\frac{1}{2}{\left(x-\hat{x}\right)}^{⊤}H\left(\hat{x}\right)\left(x-\hat{x}\right), \end{array}$$
where $H\left(\hat{x}\right)\in R^{4\times 4}$ is the approximate Hessian of V(x) evaluated at $\hat{x}$ according to (Equation 34).

We make the assumption that the uncertainty can be captured by a Gaussian distribution. Given the estimate $\hat{x}$ and the covariance matrix $P_{x}$, the probability that *x* is the true parameter vector is given by the probability density function (PDF)
(55)$$\begin{array}{cc} p(x|\hat{x},P_{x}) & =N\left(\hat{x},P_{x}\right)=\frac{1}{\sqrt{{\left(2\pi \right)}^{4}\left|P_{x}\right|}}\exp\left(-\frac{1}{2}{\left(x-\hat{x}\right)}^{⊤}P_{x}^{-1}\left(x-\hat{x}\right)\right). \end{array}$$

This is the same as assuming the estimation errors $x-\hat{x}$ to be zero-mean Gaussian with covariance $P_{x}$. We are interested in finding $P_{x}$ to quantify the uncertainty of estimates. We now consider the negative log-likelihood of this PDF
(56)$$\begin{array}{cc} -\log p(x|\hat{x},P_{x}) & =\log\left(\sqrt{{\left(2\pi \right)}^{4}\left|P_{x}\right|}\right)+\frac{1}{2}{\left(x-\hat{x}\right)}^{⊤}P_{x}^{-1}\left(x-\hat{x}\right). \end{array}$$

Note the similarities to $\hat{V}\left(x\right)$ in (Equation 54). If (Equation 54) is a good local approximation of the cost function and our estimator is unbiased, the distribution of the estimation errors $x-\hat{x}$ will be asymptotically (N→∞) zero-mean Gaussian with covariance matrix [37]
(57)$$\begin{array}{cc} P_{x} & \approx {\left(J_{s}{\left(\hat{x}\right)}^{⊤}J_{s}\left(\hat{x}\right)\right)}^{-1}, \end{array}$$
where $J_{s}\left(\hat{x}\right)$ is Jacobian from (Equation 31) where the partial derivatives of the gyroscope and acceleration residuals have been scaled by $1/\mathrm{std}(e_{\omega }\left(k,\hat{x}\right))$ and $1/\mathrm{std}(e_{a}\left(k,\hat{x}\right))$, respectively. Here, std(e(k,x)) denotes the sample standard deviation of the residuals
(58)$$\begin{array}{cc} \mathrm{std}(e(k,x)) & =\sqrt{\frac{1}{N-1}\sum_{k=1}^{N}{\left(e\left(k,x\right)-\frac{1}{N}\sum_{k=1}^{N}e\left(k,x\right)\right)}^{2}}. \end{array}$$

We want to measure the uncertainty in terms of angular deviation
(59)$$\begin{array}{c} \mathrm{AD}\left(v_{1},v_{2}\right)=\cos^{-1}\left(\frac{v_{1}^{⊤}v_{2}}{∥v_{1}∥∥v_{2}∥}\right), \end{array}$$
where $v_{1}$ and $v_{2}$ are vectors of the same dimension, $\mathrm{AD}(v_{1},v_{2})$ returns the positive angle between the two vectors. Let
(60)$$\begin{array}{cc} z & =h\left(x\right)=\left[\begin{array}{c} \mathrm{AD}(j_{1}\left(x\right),j_{1}\left(\hat{x}\right))\\ \mathrm{AD}(j_{2}\left(x\right),j_{2}\left(\hat{x}\right)) \end{array}\right], \end{array}$$
(61)$$\begin{array}{cc} x_{i} & ={\left[\begin{array}{cc} \theta_{i} & \phi_{i} \end{array}\right]}^{⊤}, \end{array}$$
then we want to find the probability distribution of p(z) or its first two moments (mean $\mu_{z}$ and covariance matrix $P_{z}$).

We use a Monte Carlo method to estimate the mean $\mu_{z}$ and covariance $P_{z}$ [38]    
(62)$$\begin{array}{cc} x_{l} & ∼N(\mu_{x},P_{x}),\;l=1,\ldots ,L \end{array}$$
(63)$$\begin{array}{cc} z_{l} & =h(x_{l}) \end{array}$$
(64)$$\begin{array}{cc} \mu_{z} & =\frac{1}{L}\sum_{l=1}^{L}z_{l} \end{array}$$
(65)$$\begin{array}{cc} P_{z} & =\frac{1}{L-1}\sum_{l=1}^{L}\left(z_{l}-\mu_{z}\right){\left(z_{l}-\mu_{z}\right)}^{⊤}, \end{array}$$
where we let $\mu_{x}=\hat{x}$, $P_{x}$ is obtained as in (Equation 57) and $h(x_{l})$ is given by (Equation 60). The covariance matrix $P_{z}$ is estimated by the unbiased sample covariance estimator, hence the division by L−1.

The metric we will be using to determine local uncertainty is the mean plus two standard deviations, $\mu_{z}+2\sigma_{z}$, where $\sigma_{z}=\sqrt{\mathrm{diag}(P_{z})}$.

### 6.2. Global Uncertainty

The cost function V(x) may have multiple local minima. In the case that the local minima correspond to either the correct or the wrong sign pairing of $j_{1}$ and $j_{2}$, we can find the correct one by comparing minima located near the opposite sign of either $j_{1}$ or $j_{2}$. If these minima are not distinctly different in terms of the values of V(x), we expect the estimates to have the correct sign half of the times our method finds a solution given that the initial estimates are uniformly spread over the parameter space. Furthermore, in the case where our data has little information about *j*, there may be other local minima that corresponds to wrong solutions. Wrong local minima can still have low local uncertainty, meaning that if our estimates are initialized near them, it is likely that wrong solutions are found. Therefore, to be confident that the method has found the global minimum, we need to solve the optimization problem multiple times with different initial estimates and compare the angular deviations of the sequential estimates.

We compute estimates ${\hat{j}}_{i}^{\left(t\right)}$ for t=1,2,…,T as
(66)$$\begin{array}{cc} {\hat{j}}^{\left(t\right)} & =\left\{\begin{array}{cc} \hat{j}, & \;t=1\\ {arg min}_{j\in \{+\hat{j},-\hat{j}\}}\min_{i\in \{1,2\}}AD\left(j_{i},{\hat{j}}_{i}^{\left(t-1\right)}\right), & \;t>1 \end{array}\right., \end{array}$$
where ${\hat{j}}_{i}^{\left(t\right)}$ is chosen as either $\pm \hat{j}$, such that one of the two estimated joint axes ${\hat{j}}_{i}$ always has the sign that is most consistent with its previous estimate. Note that this only forces either ${\hat{j}}_{1}$ or ${\hat{j}}_{2}$ to be consistent with the previous estimate, whereas the other one may still be inconsistent. We then consider the maximum sequential angular deviation as our metric for whether the estimate at time *t* corresponds to the same minimum as the estimate at time t−1
(67)$$\begin{array}{c} \mathrm{SEQAD}\left(t\right)=\left\{\begin{array}{cc} 180^{\circ }, & \;t\leq 1\\ \max_{i\in \{1,2\}}\mathrm{AD}\left({\hat{j}}_{i}^{\left(t\right)},{\hat{j}}_{i}^{\left(t-1\right)}\right), & \;t>1 \end{array}\right.. \end{array}$$

The SEQAD(t) metric corresponds to the angular deviation of the joint axis estimate that is most inconsistent with its previous estimate. Consecutive estimates will differ when there is no clear and consistent global minimum. Therefore, if we observe that SEQAD(t)→0 as *t* increases, we can be more certain that the local minimum found by our solver corresponds to a global minimum.

### 6.3. Identifying Estimates with Acceptable Uncertainty

Suppose that we receive data sequentially, i.e., we obtain $D^{N(t)}=\left\{y_{\omega }^{N(t)},y_{a}^{N(t)}\right\},\;\forall t\in \left\{0,\ldots ,T\right\}$, the sets of N(t) gyroscope and N(t) accelerometer measurements that have been recorded from time t=0 to *t*. If we use sample selection according to Algorithms 2–3, then $N\left(t\right)\leq N_{\max},\forall t$. For each $D^{N(t)}$ we obtain an estimate $\hat{j}=j\left(\hat{x}\right)$ by solving the optimization problem (Equation 22). Furthermore, we will select the estimate associated with time *t* to be ${\hat{j}}^{\left(t\right)}$ as in (Equation 66), such that either ${\hat{j}}_{1}$ or ${\hat{j}}_{2}$ is consistent with the sign of the previous estimate.

We now want to assess if ${\hat{j}}^{\left(t\right)}$ has acceptable uncertainty. Let $E_{\max}$ denote the maximum uncertainty that we accept. We use the following two criteria to determine if the local and global uncertainty is sufficiently small

We require that $\mu_{z}+2\sigma_{z}<E_{\max}$, where $\mu_{z}$ and $\sigma_{z}$ are obtained from (Equation 64) and (Equation 65) through the procedure described in Section 6.1.We require that the sequential angular deviations given by (Equation 67) satisfy $\mathrm{SEQAD}\left(t\right)<E_{\max}$ for a minimum of $n_{\min}$ consecutive estimates, that were randomly initialized uniformly over the parameter space. This is equivalent to
(68)$$\begin{array}{c} \max_{t^{\prime }\in \left[t-n_{\min}+1,t\right]}\left(\mathrm{SEQAD}(t^{\prime })\right)<E_{\max}. \end{array}$$

We summarize the method for selecting an estimate ${\hat{j}}^{\left(t\right)}$ of acceptable uncertainty in Algorithm 4.
**Algorithm 4** Identifying an estimate of acceptable uncertainty**Require:** Data $D^{N(t)}=\left\{y_{\omega }^{N(t)},y_{a}^{N(t)}\right\},\;\forall t\in \left\{1,\ldots ,T\right\}$, number of Monte Carlo samples *L*, maximum acceptable uncertainty $E_{\max}$, threshold for minimum number of sequential estimates with acceptable deviation $n_{\min}$.
1:n←02:**for** t∈{1,…,T} **do**3:    Obtain an estimate $\hat{j}=j\left(\hat{x}\right)$ by solving the optimization problem (Equation 22) using the data $D^{N(t)}$ and Algorithm 1.4:    Obtain ${\hat{j}}^{\left(t\right)}$ from (Equation 66).5:    Compute the covariance matrix $P_{x}$ according to (Equation 57).6:    Compute $\mu_{z}$ and $P_{z}$ according to the Monte Carlo method (Equation 62)–(Equation 65).7:    Compute $\mathrm{SEQAD}\left(t^{\prime }\right),\;\forall t^{\prime }\in \left(t-n_{\min}+1,t\right)$ as in (Equation 67).8:    **if**
$\mu_{z}+2\sigma <E_{\max}$ AND $\max\left\{\mathrm{SEQAD}\left(t^{\prime }\right)\right\}<E_{\max}$ **then**9:        **return**
${\hat{j}}^{\left(t\right)}$.10:    **end if**11:**end for**

## 7. Experiment

### 7.1. Data Acquisition

Data were collected from a 3D printed hinge joint system [39] with one wireless IMU (Xsens MTw) attached to each segment; see Figure 3. The sampling rate was set to 50Hz for both IMUs. The operating ranges of the IMUs were ±160 m/s^2^ for the accelerometers and ±21 rad/s for the gyroscopes. The IMUs were attached in sockets such that the joint axis was parallel with the positive y-axes of the sensors and both pointing in the same direction (same sign), that is
(69)$$\begin{array}{cc} j_{1} & =j_{2}={\left[\begin{array}{ccc} 0 & 1 & 0 \end{array}\right]}^{⊤}. \end{array}$$

The data consist of 14 recorded motions, listed in order below

Stationary system;Free rotation, stiff joint, free joint axis;Sequential rotation, horizontal joint axis;Sequential rotation, tilting joint axis;Simultaneous planar rotation, horizontal joint axis;Simultaneous planar rotation, tilting joint axis;Simultaneous free rotation, free joint axis;[8–14] Same motions as 1–7, respectively, but with faster rotations.

The recorded angular velocity magnitudes for these motions are shown in Figure 4. For the sequential rotation, Segment 1 was always rotated first while Segment 2 was stationary, which was followed by the converse motion. Horizontal joint axis means that the joint axis was aligned to be approximately orthogonal to the gravitational acceleration vector. For the tilting joint axis case, the angle between the joint axis and the gravitation acceleration vector was maintained at $\approx 45^{\circ }$ for the duration of the motion. For the free joint axis motions, the joint axis was not constrained to any particular orientation, but rotated freely in space. The hinge joint system was equipped with a screw, which when tightened prevented independent rotation of the two segments. The screw was tightened when the system was stationary and during the stiff joint rotations. Measurements of the transitions from one motion to another were removed from the recorded data, such that only the specified motions of interest could be isolated. The first set of stationary data was used to estimate the gyroscope bias $b_{\omega }$ in (Equation 3), which was then subtracted from all subsequent measurements.

Because the optimization problem (Equation 22) is formulated based on the kinematics at each sample time, and does not contain dynamics, we are allowed to shuffle around the measurements in our data set. Using the 14 different motions, four scenarios were designed where the motions appeared in different sequences. The sequences of motion for the different scenarios were

1,2,3,4,5,6,7,8,9,10,11,12,13,14;$1,3^{*},10^{*},8,2,9,3,10,4,11,5,12,6,13,7,14$;^*^ only 500 samples (10 seconds) from motions 3 and 10, which contain motion in only Segment 1.$6^{*},1,8,2,9$;^*^ only 1000 samples from motion 6.$1,8,2^{†},9^{†},6^{*},2^{†},9^{†}$.^*^ only 1000 samples (20 seconds) from motion 6.^†^ samples divided in half.

Scenario 1 is the original sequence in which the motions were recorded. Scenario 2 starts with the sensors being stationary, then there is motion in only Segment 1, after which the system alternates between the slower and faster motions, starting with non-informative stiff-joint motions. For this scenario we expect to have good estimates of $j_{1}$ before we have any excitation in Segment 2. Scenario 3 has early excitation of both segments followed by measurements from a stationary system and non-informative stiff joint motion and Scenario 4 has the converse case where the excitation comes late in the sequence. Scenarios 3 and 4 are also designed to contain more non-informative motions, as the only informative motion is contained in the 1000 samples (20 seconds) from motion 6.

### 7.2. Evaluating Robustness of the Residual Weighting

To experimentally evaluate the robustness of the proposed method for different weights $w_{\omega }$ and $w_{a}$, we estimated the joint axis using data from motions 3 to 7 and 10 to 14. Data from a stationary system and rotations with a stiff joint were not used in these evaluations since the joint axis is not identifiable for these motions. The weights were chosen as
(70)$$\begin{array}{cc} w_{\omega } & =\sqrt{w_{0}},\;w_{a}=\frac{1}{\sqrt{w_{0}}}, \end{array}$$
where we let $w_{0}=\frac{w_{\omega }}{w_{a}}$ determine the relative weighting of the residuals $e_{\omega }$ and $e_{a}$. We estimated the joint axis for 100 different values of $w_{0}$, which had a logarithmic distribution on the interval $\left(10^{-3},10^{10}\right)$. For all different motions and for each value of $w_{0}$ the initial estimates of *x* were selected deterministically such that all possible sign pairings $\left(\pm j_{1},\pm j_{2}\right)$ and $\left(\pm j_{1},\mp j_{2}\right)$ were selected equally many times. The initial estimates for $j_{1}$ and $j_{2}$ were selected from a grid on the unit-sphere of $R^{3}$ with 6 grid points the positive and negative axes. With all possible combinations for $j_{1}$ and $j_{2}$ this resulted in M=36 different initial conditions for the optimization algorithm. Here we use the root-mean-square angular error (RMSAE) for both joint axes as the metric to evaluate performance
(71)$$\begin{array}{cc} \mathrm{RMSAE} & =\sqrt{\frac{1}{2M}\sum_{k=1}^{M}\mathrm{AD}{\left(j_{1},j\left({\hat{x}}_{1}^{\left(k\right)}\right)\right)}^{2}+\mathrm{AD}{\left(j_{2},j\left({\hat{x}}_{2}^{\left(k\right)}\right)\right)}^{2}}, \end{array}$$
where AD is the angular deviation metric given by (Equation 59), and here we let the superscript *k* denote the estimates obtained from different initializations. Since we consider $\left(\pm j_{1},\pm j_{2}\right)$ to be correct sign pairings of the joint axis, we select the sign of ${\hat{j}}_{1}$ which has the lowest AD. If, as a result of this, ${\hat{j}}_{1}$ changes sign, the sign of ${\hat{j}}_{2}$ is also changed. This way AD for ${\hat{j}}_{1}$ will always be $\leq 90^{\circ }$ whereas the AD for ${\hat{j}}_{2}$ can be up to $180^{\circ }$, which corresponds to an error in the sign pairing.

### 7.3. Evaluating Sample Selection

To evaluate the proposed sample selection in Algorithms 2–3, we use the data according to the four scenarios specified in Section 7.1. Starting with the first second of recorded motion and incrementally adding subsequent data in small batches of one second at a time. That is, we receive sequential batches of data $D^{N(t)},\forall t\in \left\{1,\ldots ,T\right\}$ where *T* is the duration of the scenario.

We compute one estimate ${\hat{j}}_{i}^{\left(t\right)}$ for each $D^{N(t)}$. For each new batch, the joint axis is estimated again starting from a new initial estimate (i.e., no warm-start of the optimization method), which is randomly selected from a uniform distribution. The reason for this is that we also want to evaluate if the estimates are consistent over time, regardless of initialization of the optimization method. The relative weighting of the residuals (Equation 70) was set to $w_{0}=50$.

We compare the method when the proposed sample selection is used to the case where all available data is used ($N_{\max}=N\left(t\right)$) for estimation for all four scenarios. When using the sample selection in Algorithms 2–3, the maximum sample sizes of $N_{\max}\in \left\{1000,500,250,125\right\}$ were compared.

Other than *N*, the other user chosen parameters for the sample selection is related to the angular rate energy penalty (Equation 45). The window size of n=21 samples was used, which means the average energy is computed for 0.42s of motion for our sensors. The threshold used to determine if the accelerometer is stationary in Algorithm 3, was set to $E_{\mathrm{th}}=1$rad^2^s^−2^. This is around 10 times higher than the threshold used for the angular rate energy detector suggested for zero-velocity detection in human gait [40]. Measurements that we discard in line 2 of Algorithm 3 are therefore likely to be of significant motion.

### 7.4. Evaluating Uncertainty Quantification

To evaluate the efficacy of the proposed uncertainty quantification, we will use the same sequential batches of data for all four scenarios $D^{N(t)},\forall t\in \left\{1,\ldots ,T\right\}$ as in Section 7.3, but with a fixed maximum number of samples N=1000 chosen by Algorithms 2–3. Similarly to the procedure used to evaluate the sample selection method one estimate ${\hat{j}}_{i}^{\left(t\right)}$ is obtained for each new batch, and initial estimates are independently randomized from a uniform distribution over the parameter space at each *t*. The relative weighting of the residuals (Equation 70) was set to $w_{0}=50$.

Here we use Algorithm 4, which returns an estimate $\hat{j}$, when the local and global criteria indicate that the uncertainty is acceptable. This requires the user to choose the threshold for acceptable uncertainty, $E_{\max}$, and the minimum number of sequential estimates that should have angular deviations below this threshold, $n_{\min}$. For our evaluation we choose to set $E_{\max}=3^{\circ }$, and $n_{\min}\in \left\{3,10\right\}$. Algorithm 4 is then deemed to be successful if $\mathrm{AD}\left(j_{i},{\hat{j}}_{i}\right)\leq E_{\max}$. This procedure is repeated 100 times for each scenario, with different randomized initial estimates each time.

### 7.5. Evaluating Robustness to Sensor Bias

We evaluated the robustness of the complete method, which includes Algorithms 1–4, to measurement bias. The measurement bias refers to $b_{a}$ and $b_{\omega }$ in the measurement models (Equation 1)–(Equation 3). In the other evaluations presented here, we have compensated for gyroscope bias by estimating $b_{\omega }$ from the initial stationary data (Motion 1) and subtracting this bias from the subsequent measurements. We have not compensated for any accelerometer bias since it cannot be estimated from only one stationary position of the sensor. In this section, we will study the effect of sensor biases by adding artificially generated biases to both the previously bias-compensated gyroscope measurements and to the accelerometer measurements. These artificial biases have fixed magnitudes $∥b_{a}∥=1$ m/s^2^ and $∥b_{\omega }∥=1^{\circ }/$s, but their directions are randomized by generating random unit vectors. To evaluate the effect of the added artificial bias, M=100 estimation runs are performed for all four scenarios, with and without the added artificial bias. The artificial biases are first added to the measurements, then the proposed method is applied as described in Section 7.4. Here, we set $N_{\max}=500$, $E_{\max}=1^{\circ }$ and $n_{\min}=10$, other parameters are the same as in previous sections. We will use the RMSAE metric (Equation 71) and the maximum angular error (MAXAE)
(72)$$\begin{array}{cc} \mathrm{MAXAE} & =\max_{1\leq k\leq M}\left\{\mathrm{AD}\left(j_{1},j\left({\hat{x}}_{1}^{\left(k\right)}\right)\right),\mathrm{AD}\left(j_{2},j\left({\hat{x}}_{2}^{\left(k\right)}\right)\right)\right\}, \end{array}$$
to evaluate the performance of the method across all M=100 estimation rounds for all scenarios.

## 8. Results

### 8.1. Robustness

The RMSAE for the different motions and weights $w_{0}$ are shown in Figure 5. Here, estimation is done separately and using all samples for each motion, i.e., without sample selection. The optimal choice for $w_{0}$ appears to be in the interval of $\left(10^{1},10^{5}\right)$, where the errors are small for all motions.

### 8.2. Sample Selection

The angular errors for ${\hat{j}}_{1}$ and ${\hat{j}}_{2}$ obtained from testing the proposed sample selection as described in Section 7.3, are shown in Figure 6. From these results, we can compare the use of Algorithms 2–3 for different sample sizes $N_{\max}$. This includes the case of $N_{\max}=N\left(t\right)$, where N(t) corresponds to making use of all samples that have been observed up to an integer *t* number of seconds.

Figure 7 shows which samples were selected for $N_{\max}=1000$, for Scenarios 1 and 2 at the times given by the vertical axes.

### 8.3. Uncertainty Quantification

With the parameter $n_{\min}=10$, the final errors were below $E_{\max}=3^{\circ }$ for all 100 estimation rounds for all four scenarios, meaning the estimates obtained from Algorithm 4 were acceptable 100% of the time. With $n_{\min}=3$ and the same $E_{\max}$, the estimates were acceptable 8% of the time for Scenario 1, 81% of the time for Scenario 2, 100% of the time for Scenario 3 and 0% of the time for Scenario 4. Figure 8 compares the local and global uncertainty metrics to the angular errors of a single estimation round for each scenario and shows when Algorithm 4 accepted an estimate $\hat{j}$ for $n_{\min}=3$ (leftmost vertical dashed lines) and for $n_{\min}=10$ (rightmost vertical dashed lines).

### 8.4. Robustness to Sensor Bias

The resulting RMSAE and MAXAE for the M=100 estimation rounds with and without added artificial bias are shown for all four scenarios in Table 1. Without the added artificial biases, the errors were at most $2.16^{\circ }$, and with the added artificial biases of magnitudes $∥b_{a}∥=1$ m/s^2^ and $∥b_{\omega }∥=1^{\circ }/$s the errors were at most $4.84^{\circ }$.

## 9. Discussion

### 9.1. The Method Is Not Sensitive to the Relative Weighting $w_{0}$

The parameter $w_{0}$, which is defined from (Equation 70), controls the relative weighting of the residuals $e_{\omega }$ and $e_{a}$. As $w_{0}$ increases, the relative weighting of the gyroscope residual is increased. As we see in Figure 5, the optimal choice of $w_{0}$ for most motions in terms of RMSAE (Equation 71), is somewhere in the large range between 10 and $10^{5}$. The errors are also small ($<3^{\circ }$) for $w_{0}<10$ for the slower planar motions (3–6), which shows that the acceleration information can be reliable for these motions. However, some larger errors can be observed for small $w_{0}$ for the faster planar motion 12 and the errors are also significantly larger for the free axis rotations (motions 7 and 14), which can be explained by the fact that these motions violate the acceleration constraint, meaning that the r.h.s. of (Equation 17) is nonzero.

Since we can select any $w_{0}$ from within such a large interval and still obtain similar performance, our method is not sensitive to the relative weighting of the residuals. It makes sense that $w_{0}>10$, since the angular velocities, measured in rad/s, have smaller magnitudes than the accelerations, that typically fluctuate around 9.82 m/s^2^ due to the gravitational acceleration. Furthermore, the angular velocity constraint always holds for a rigid hinge joint system. Hence, we expect the angular velocity information to be more reliable. The method is robust for larger $w_{0}$ up to $10^{5}$ where the RMSAE become large for motions 3 and 10. This large increase in RMSAE occurs when the acceleration residual becomes numerically indistinguishable to the tolerance of the optimization algorithm, and it becomes more likely that the method selects an estimate which corresponds to the wrong sign pairing. Therefore, as $w_{0}$ increases we see the RMSAE approach $90^{\circ }$ as the AD for ${\hat{j}}_{1}$ is still small but the probability of selecting $\pm {\hat{j}}_{2}$ is approaching 0.5, meaning that approximately half of the estimates will have the wrong sign pairing. This can also depend on the numerical tolerance and stopping criteria of the optimization method, since a global minimum corresponding to the correct sign pairing might not be significantly different from other local minima that correspond to the wrong sign pairing.

### 9.2. Sample Selection Offers Substantial Benefits

From the results shown in Figure 6 we see that we can achieve similar, and in some cases even better performance by selecting relatively few measurements to use for estimation out of all N(t) measurements that have been observed up to time *t*. For Scenario 1, $N_{\max}\in \left\{1000,500,250\right\}$ have angular errors within $0.5^{\circ }$ and N=125 have errors within $1^{\circ }$ from the case with $N_{\max}=N\left(t\right)$.

In Scenario 2, the errors for ${\hat{j}}_{2}$ drop below $2^{\circ }$ at t=85s for the methods using sample selection, but it takes until t=135s for the method where $N_{\max}=N\left(t\right)$ to stay consistently below $2^{\circ }$. However, the method with N=125 again shows a slightly larger deviation from the others, with some momentary spikes in error around t=200s and t=300s. Note that Scenario 2 is designed to have no independent rotation of Segment 2 until t=255s. So the only information about $j_{2}$ until then has to come from the accelerometer. This shows that carefully selecting accelerometer samples according to Algorithm 3 is beneficial, especially if angular velocity information is missing. Comparing the results from Scenarios 1 and 2, the final errors are very similar, indicating that the methods are not sensitive to the sequence of motions.

Scenarios 3 and 4 represent challenging cases where only a small minority of samples contain motion with independent rotation (only 20s of motion 6). In Scenario 3, we note that the final error for $N_{\max}=N\left(t\right)$ is significantly larger than the cases with $N_{\max}\in \left\{1000,500,250\right\}$, and for $N_{\max}=125$ the final error for ${\hat{j}}_{2}$ is at the same level as $N_{\max}=N\left(t\right)$. Scenario 4 has a similar performance in terms of final errors. However, Scenario 4 does not have any motions with independent rotations of the segments until around t=180s. The only information about the joint axis until that point comes from the accelerometer, in Motions 1, 8, 2 and 9. The errors start to decrease around t=150s when data from Motion 9 comes in, but do not settle until after Motion 6. The large fluctuations in errors we see for $N_{\max}=1000$ during Motion 9 indicate that there are still at least two local minima corresponding to the wrong joint axis at this point. Errors for $N_{\max}<1000$ are smaller during motion 9, but still vary between $5^{\circ }$ and $20^{\circ }$.

Using Algorithms 2–3 is therefore beneficial, not only for reducing the computational complexity of the optimization problem, but it can even improve the performance in situations where gyroscope information is limited. However, judging by Scenario 4 in particular, $N_{\max}\geq 1000$ appears to be the best choice in terms of overall performance. Even then, $N_{\max}=1000$ is only a small fraction of the total number of measurements. With a sample period of 0.02s, we have that N(t=700)=35000 and N(t=250)=12500.

Figure 7 shows the samples that were selected over time from Scenarios 1 and 2 with $N_{\max}=1000$ and which motions these samples come from. For both scenarios, we see that gyroscope samples from non-informative motions 1,2,8,9 are all deselected by the end. Samples from these motions are only kept until enough samples from informative motions have been parsed by the algorithm. For the accelerometer, we see that samples from stationary sensors are preferred since many samples of motions 1 and 8 are kept, which is in line with the penalty we give samples based on the angular rate energy. It is also important that samples from other motions are selected since the criterion for identifiability requires a strong separation between the nullspace and the column space of the matrix *A* given by (Equation 27). Had the selection criterion of the accelerometer only been based on the angular rate energy, we would risk ending up in the situation where all samples are selected from the same stationary position, in which case all rows of *A* are linearly dependent. Lines 5–10 in Algorithm 3 prevent this by removing the worst samples that are coherent with the right-singular vector of the largest singular value. This can be thought of as allocating space in the *A* matrix for novel information by removing redundant information.

### 9.3. Reliability of the Proposed Uncertainty Quantification

We obtained reliable estimates with errors below that of the maximum acceptable error $E_{\max}=3^{\circ }$, 100% of the time when the parameter $n_{\min}=10$. However, estimates were not reliable for $n_{\min}=3$, where the results were particularly bad for Scenario 1, with 8% of estimates of acceptable error and for Scenario 4 with 0% of estimates of acceptable error. Both of these scenarios contained no informative motions in the beginning, and we found that the estimates that were returned often had not used any batch of informative data for estimation because the criteria for local and global uncertainty were satisfied prematurely by Algorithm 4.

In Figure 8 it can be seen that local uncertainty metric $\mu_{z}+2\sigma_{z}$ can be below $E_{\max}$ (horizontal dashed lines) while the actual angular error fluctuates between values below and above $E_{\max}$. This occurs when there exist multiple other local minima than those corresponding to the true joint axis. Furthermore, Figure 8 shows how the SEQAD remains large as the angular errors fluctuate in the same way as the angular errors. Interestingly, Scenario 2 appears to fluctuate between one correct and one wrong local minimum between t=66s and t=82s. If we assume that the probability of finding the correct local minimum is 0.5, having $n_{\min}=3$ that means that the probability of ending up in the wrong local minimum $n_{\min}$ times in a row is $0.5^{n_{\min}}=12.5\%$. This matches well with the results obtained for Scenario 2, where 88% of the estimates were acceptable for $n_{\min}=3$.

For Scenarios 1 and 4, where the results were significantly worse for $n_{\min}=3$, it appears that wrong local minima were dominating. These two scenarios have sequences of stiff-joint motion before any informative motions are observed, which can explain why wrong local minima were found more frequently. Scenario 3, which had informative motion in the beginning did not have this issue, and hence 100% of the estimates were acceptable even for $n_{\min}=3$.

We can conclude that setting the parameter $n_{\min}$ sufficiently large is important for fully capturing the global uncertainty. Sequential data dominated by non-informative motions in the beginning are more sensitive to the choice of $n_{\min}$. The results showed that Algorithm 4 successfully identified all of the estimates that satisfied the accuracy criteria $E_{\max}=3^{\circ }$ when $n_{\min}=10$. Here, this corresponds to 10 consecutive estimates (computed once per second), that differed by $3^{\circ }$ at most.

### 9.4. The Method Is Robust to Realistic and Uncompensated Sensor Bias

As shown in Table 1, even with added artificial biases of relatively large magnitudes $∥b_{a}∥=1$m/s^2^ and $∥b_{\omega }∥=1^{\circ }/$s, the errors were at most $4.84^{\circ }$ across all M=100 estimation runs for all four scenarios. The average errors in terms of RMSAE were less than $2^{\circ }$ even with the added artificial biases. As a comparison, the IMUs used in our experiments had bias magnitudes in the order of $∥b_{a}∥=0.1\;$ m/s^2^ and $∥b_{\omega }∥=0.5^{\circ }/$s, so the artificial biases were significantly larger. This shows that the method is robust to sensor biases of at least these magnitudes. However, we had to lower the threshold $E_{\max}$ from $3^{\circ }$ to $1^{\circ }$ to achieve this. This means that Algorithm 4 will be more conservative in selecting an estimate. With added artificial bias and $E_{\max}=3^{\circ }$, the method would sometimes terminate prematurely, when no informative motion had been observed because a global minimum that satisfied this threshold value was found. Therefore, lowering $E_{\max}$ was required to achieve robustness to the added artificial biases. It is therefore still highly recommend that pre-calibration of the biases is performed when possible. If bias drift is significant enough to exceed the magnitudes tested here across the duration of the experiment, it is advised to use a method that allows for online compensation of biases alongside the proposed method. Lowering $E_{\max}$ is only an optional measure one would take in the unusual case where late bias occurs and is not compensated for.

## 10. Conclusions

We have proposed a method which facilitates plug-and-play sensor-to-segment calibration for two IMUs attached to the segments of a hinge joint system. The method identifies the direction of the joint axis *j* in the intrinsic reference frames of each sensor, thus providing the user with information about the sensors’ orientation with respect to the joint. Accurate sensor-to-segment calibration is crucial for tracking the motion of the segments.

The method was experimentally validated on data collected from a mechanical joint, which performed a wide range of motions with different identifiability properties. As soon as sufficiently informative data was available, the method achieved a sensor-to-segment calibration accuracy in the order of $2^{\circ }$, assessed as the angular deviation from the ground truth of the joint axis.

The proposed method includes the following features that were evaluated separately using the experimental data:Gyroscope and accelerometer information are weighted and combined, which makes the joint axis identifiable for a wider range of different motions. Experimental evaluation showed that the method is not sensitive to the weighting parameters, and that it performs comparably well for a wide range of different motions across a large interval of weights.A method to select a smaller subset of samples to use from a long sequence of recorded motion is proposed. Samples are selected from motions that yield identifiability, and measurements of non-informative motions are automatically discarded. The experimental evaluation showed that using between 125 and 1000 samples can achieve similar and in some cases even better performance than using all available samples collected from a long sequence of motions. Sample selection was shown to be particularly beneficial when data consisted of more non-informative than informative motions. Furthermore, using less samples for estimation reduces the computational complexity of the estimation.A method to quantify local and global uncertainty properties of sequential estimates, which provides the user with an estimate when criteria for acceptable uncertainty are met. The method successfully identified estimates that satisfied the uncertainty criteria ($E_{\max}=3^{\circ }$).

The proposed method is the first truly plug-and-play calibration method that directly enables plug-and-play motion tracking in hinge joints. For the first time, the user can simply start using the sensors instead of performing precise or sufficiently informative motion in a predefined initial time window, and the proposed method provides reliable calibration parameters as soon as possible, which immediately enable calculation of accurate motion parameters from the incoming raw data as well as from already recorded data. Regardless of the performed motion, it provides only parameters that are actually accurate, which is not guaranteed by any state of the art method. This enables the kind of truly non-restrictive and reliable motion tracking that is needed in a range of application domains including ubiquitous motion assessment to wearable biofeedback systems.

In future work, the method could be extended to different joint types and be applied to motion tracking in mechatronic and biomechanical systems. For the latter case in particular, it would be of great interest to study the reliability of the method in non-rigid systems, such as human limbs, where motion of soft tissue is significant.

## Figures and Tables

**Figure 1 sensors-20-03534-f001:**
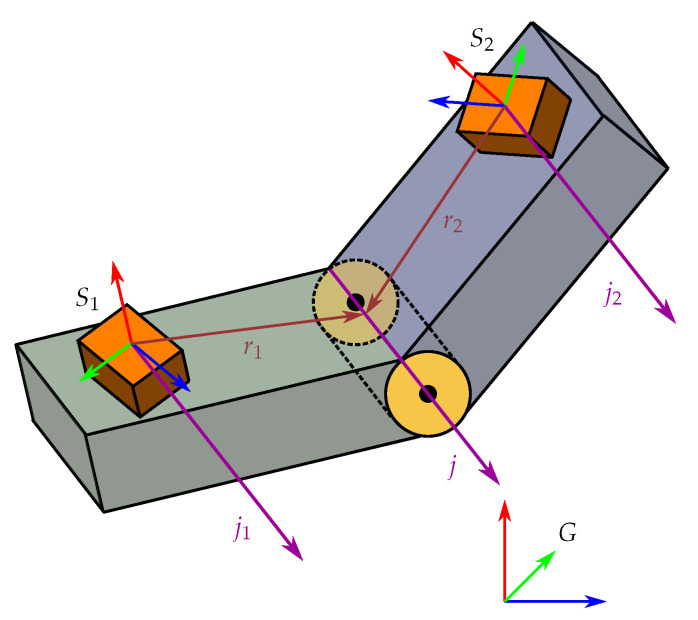
The hinge joint system that we consider. The two segments rotate independently with respect to each other only along the joint axis *j*. The sensor frames $S_{i}$ are rigidly fixed to their respective segments and their relative orientation can be described by one joint angle, that corresponds to a rotation about the joint axis. The joint axis expressed in local sensor coordinates is an important sensor-to-segment calibration parameter in joint systems with one degree of freedom (DOF).

**Figure 2 sensors-20-03534-f002:**
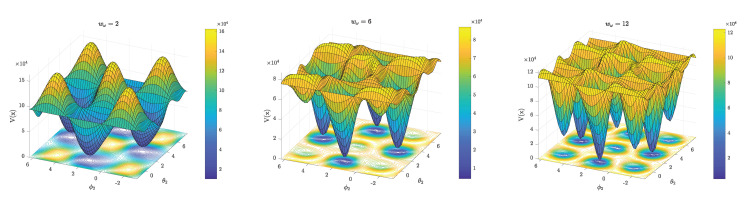
Shape of the cost function V(x) for a motion with simultaneous planar rotations of the segments. The parameters $\theta_{1}$ and $\phi_{1}$ are fixed near their true values and $\theta_{2}$ and $\phi_{2}$ are allowed to vary. From left to right, we see how the geometry changes as $w_{\omega }$ increases while $w_{a}=1$ is constant. As $w_{\omega }$ increases, new local minima appear near the locations at $\phi_{2}+\pi$ from the previously existing local minima. These new local minima correspond to the wrong sign pairing $\left(\pm j_{1},\mp j_{2}\right)$.

**Figure 3 sensors-20-03534-f003:**
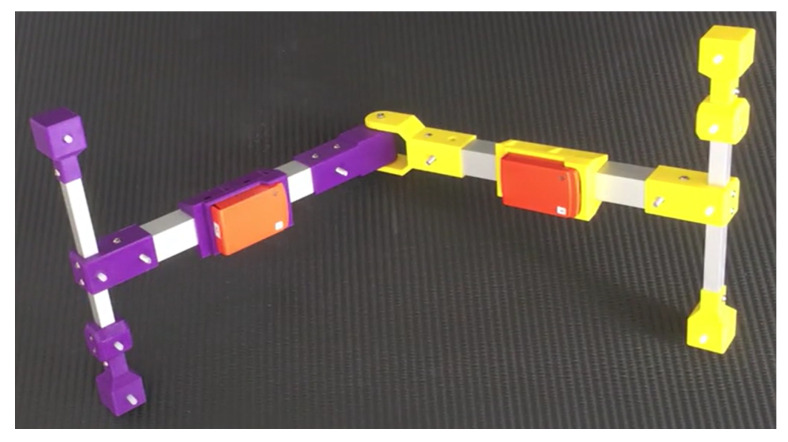
The 3D printed hinge joint system, design by Dustin Lehmann, with the two IMUs (orange boxes, 34×58 mm) attached.

**Figure 4 sensors-20-03534-f004:**
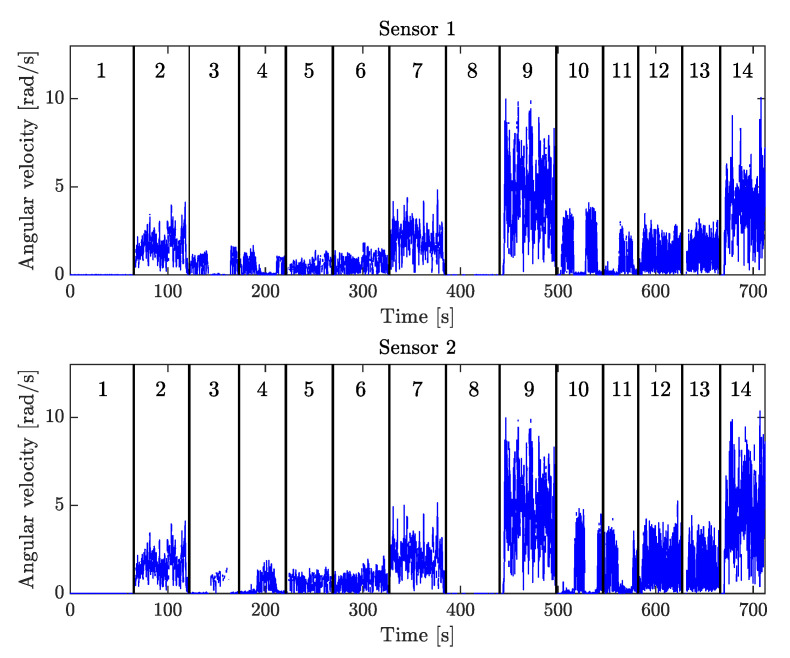
The angular velocity magnitudes for the 14 different motions that were recorded. The vertical lines and numbered sections indicate when the different motions begin and end.

**Figure 5 sensors-20-03534-f005:**
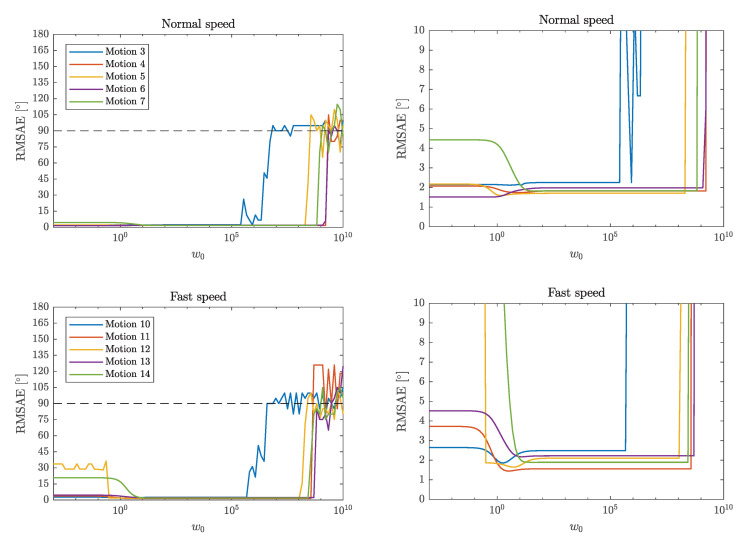
Root-mean-square angular error (RMSAE) (Equation 71) for different motions and weights $w_{0}$. Normal speed motions are shown in the top plots and faster motions are shown in the bottom plots. The plots on the right show the same results as the plots to their respective lefts, but zoomed in.

**Figure 6 sensors-20-03534-f006:**
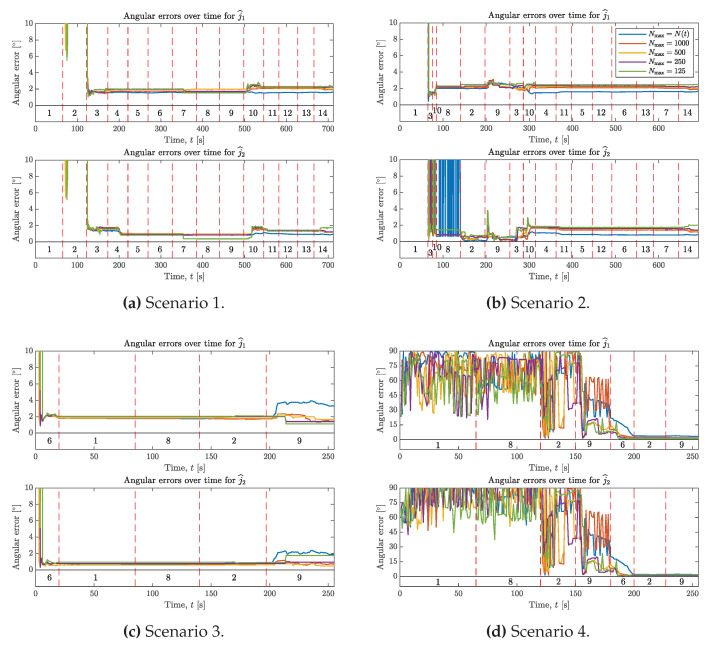
Angular errors over time for the four scenarios, see (**a**–**d**). Comparing the case $N_{\max}=N\left(t\right)$, where all samples up to time *t* are used for estimation to $N_{\max}\in \left\{1000,500,250,125\right\}$ samples being chosen by Algorithms 2–3 at each integer *t* seconds. Colored lines define these different cases as given by the legend in the top right. Vertical dashed lines and the numbers 1–14 are used to indicate from which motions (see Section 7.1) the data comes from.

**Figure 7 sensors-20-03534-f007:**
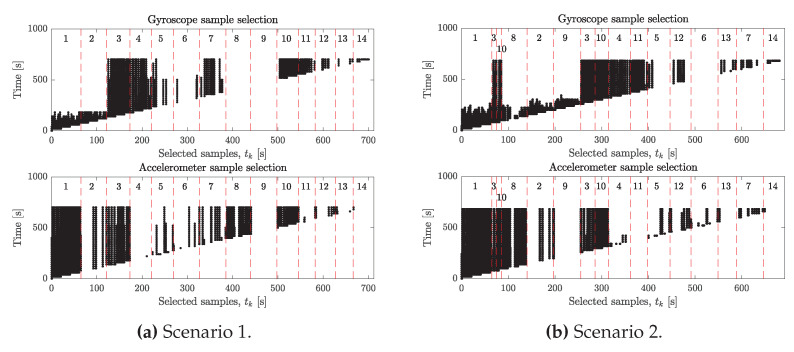
The figure shows which samples from Scenario 1 (**a**) and Scenario 2 (**b**) that were selected by Algorithms 2–3 with $N_{\max}=1000$, at the times given by the vertical axes. Black/white indicates that a sample were selected/not selected respectively. As time increases and more samples become available, we see some previously selected samples being deselected in favor of new samples that are deemed superior by the algorithms. Vertical dashed lines and the numbers 1–14 indicate from which motions (see Section 7.1) the data comes from.

**Figure 8 sensors-20-03534-f008:**
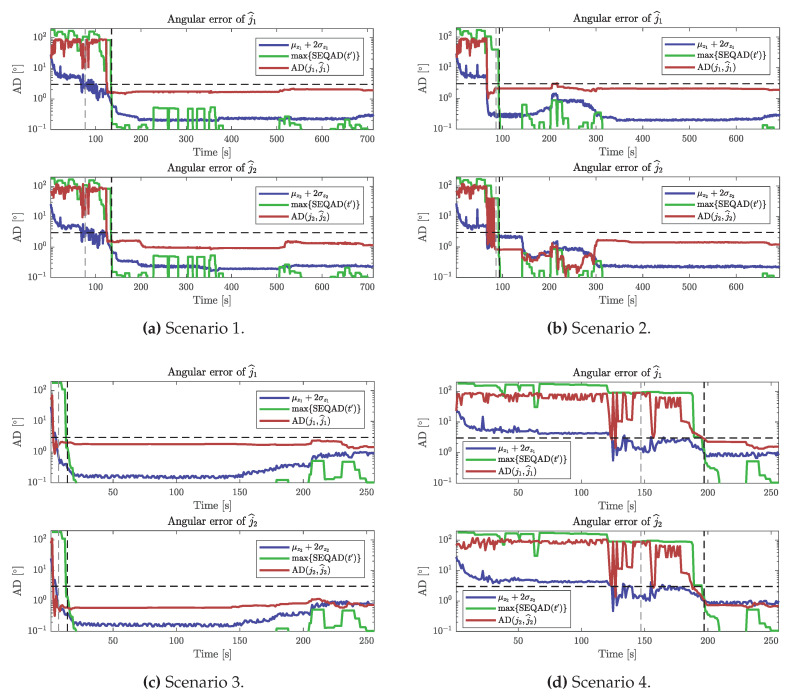
The plots shows the local and global uncertainty metrics compared to the angular errors (red) for the four scenarios, see (**a**–**d**). Local uncertainty is quantified by $\mu_{z}+2\sigma_{z}$, where $\mu_{z}$ is the estimated mean AD (Equation 64) and $\sigma_{z}$ is the standard deviation, computed from the estimated covariance matrix (Equation 65). Local uncertainty (blue) is shown for both ${\hat{j}}_{1}$ and ${\hat{j}}_{2}$ for each scenario. Global uncertainty is quantified by (Equation 68). The global uncertainty (green) with $n_{\min}=10$ is shown for each scenario. Horizontal dashed lines show the accuracy threshold $E_{\max}=3^{\circ }$. Vertical dashed lines show when estimates $\hat{j}$ were accepted by Algorithm 4. For each scenario, the leftmost vertical lines show the case of $n_{\min}=3$ and the rightmost vertical lines show the case of $n_{\min}=10$, where Algorithm 4 terminates when the estimates have reached the desired accuracy w.r.t. ground truth.

**Table 1 sensors-20-03534-t001:** Shows the RMSAE (Equation 71) and MAXAE (Equation 72) after M=100 runs with and without artificial bias for the four scenarios.

Scenario	$∥b_{a}∥$ [m/s^2^]	$∥b_{\omega }∥$ [°/s]	RMSAE [°]	MAXAE [°]
1	0	0	1.55	1.67
1	1	1	1.73	4.41
2	0	0	1.58	2.16
2	1	1	1.97	4.84
3	0	0	1.50	2.07
3	1	1	1.58	3.09
4	0	0	1.47	1.98
4	1	1	1.30	2.32

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
