# Peer review of "Robust Plug-and-Play Joint Axis Estimation Using Inertial Sensors"

_sensors, 2020, doi:10.3390/s20123534_

Round 1

Reviewer 1 Report

I would like to congratulate the authors on producing a high quality manuscript which details thoroughly the implementation of a new framework to estimate joint axis using inertial sensors.

The authors provided an extensive review of literature which highlighted the novelty of their paper. I might just ask in the introduction after this paragraph:

"In the proposed system, successful calibration no longer depends on performing certain motions in a predefined manner or time window but only on fulfilling the minimum required conditions at some point." (Lines 78-79) to provide some examples of what these minimum required conditions at some point would be.

The authors showed a through and detailed explanation of the mathematical and physics theories which support the soundness of their algorithm. They also used a cost function and optimisation technique to ensure they achieve the best joint angle estimation of a high number of samples.  The testing procedure was sound, they used a 3D printed hinge joint system with two IMU's attached for close to real-life results. Not sure if missed- Might be worth adding the properties of IMUs used such as sampling rate and operating range of the accelerometer and gyroscope.

The joint angle estimations were of high accuracy with errors in the order of only 2 degrees being achieved. The authors have presented a highly accurate plug and play system joint angle estimation system for hinge joints (1DOF). They suggest future work would be to extend this to different joint types and be applied to motion tracking in mechatronic and biomechanical systems. I believe this will be a good tool in these fields.

Author Response

We want to thank the reviewers for their feedback and comments, which has helped us in improving the quality of our manuscript. Here is how we have adressed the specific comments from Reviewer 1 in our revision:

Reviewer 1: "In the proposed system, successful calibration no longer depends on performing certain motions in a predefined manner or time window but only on fulfilling the minimum required conditions at some point." (Lines 78-79) to provide some examples of what these minimum required conditions at some point would be."

The meaning of the term "informative" and of the corresponding minimum conditions is pointed out much clearer in the introduction of the revised manuscript, where both the corresponding reference and its main results are presented. Moreover, Section 4.2 provides further details and examples for non-informative motions.

Reviewer 1: "Might be worth adding the properties of IMUs used such as sampling rate and operating range of the accelerometer and gyroscope."

The sampling rate of the IMUs is stated in the second sentence of Section 7.1. We have also added the operating ranges in the same section.

Reviewer 2 Report

This manuscript introduces a new method to identifies the direction of the joint axis in the intrinsic reference frames of two IMUs attached to the segments of the hinge.  As the author claimed that the major advantage of proposed method is to start using the sensors instead of performing precise or sufficiently rich motion in a predefined initial time window. However, it is not clear that if the sensors can be used to tracking the motion of the segments before the proposed method provides accurate calibration parameters from the "random" motion. If it is not the case, then I did not find significant improvement of this new method in practice comparing to do the calibration in an initial time window by using some pre-defined motions. Maybe the proposed algorithms for sample selections are good supplements to the state of art calibration methods but it is still not a "plug-and-play" way for joint motion detection.

Also, in the part of "Inertial measurement models", the authors use very ideally assumptions to exclude the impact of the drift of bias and scale factors, the sensor axis mis-alignment, the 1/f noises or other non-Gaussian noise and so on. However, these issues are very common in low-cost MEMS IMU which are widely used in the human body motion detection. So, as a real "plug-and-play" method, the authors should include these non ideal but intrinsic factors into their model and evaluate their impacts to the estimation uncertainty. 

Author Response

We want to thank the reviewers for their feedback and comments, which has helped us in improving the quality of our manuscript. Here is how we have adressed the specific comments from Reviewer 2 in our revision:

Reviewer 2: "However, it is not clear that if the sensors can be used to tracking the motion of the segments before the proposed method provides accurate calibration parameters from the "random" motion. If it is not the case, then I did not find significant improvement of this new method in practice comparing to do the calibration in an initial time window by using some pre-defined motions. Maybe the proposed algorithms for sample selections are good supplements to the state of art calibration methods but it is still not a "plug-and-play" way for joint motion detection."

The comment is very valuable in the sense that it makes us notice that we might not have explained the practical relevance of plug-and-play calibration well enough. IMU-based motion tracking is used in healthy subjects and in patients with motor weakness or impairments. It is used

  • for online purposes, such as biofeedback and automatic parameter adaptation in assistive devices,
  • and for offline purposes, such as ergonomic workplace assessment or performance analysis in sports.

The question whether "the sensors can be used to [assess] the motion of the segments before the proposed method provides accurate calibration parameters" must be answered with "yes". The orientation data of both sensors can be recorded from the start, and the motion between the start t_0 and the moment t_i at which the parameters become identifiable can be analyzed retrospectively as early as in the moment t_i. In practice, this means that, for example, an elderly person starts walking right after attaching the sensors, which calibrate themselves during the first ten strides and then provide biofeedback to the person if the gait was asymmetric during those first ten strides or thereafter as soon as the gait becomes asymmetric.

Obviously, when the moment t_i is reached depends on the motion the user performs. If timely calibration is desirable, the user will perform rich motions until the system indicates that the desired calibration accuracy is reached. Note the remarkable difference to existing solutions in which the user moves for a predefined "long enough" time and neither receives notification if the calibration was already successful after half of the predefined time nor receives notification if the calibration fails.

The question for "significant improvement of this new method in practice comparing to do the calibration in an initial time window by using some pre-defined motions" is addressing the very core of the value of our contribution. In all of the aforementioned applications, the proposed method yields significant improvements compared to conventional anatomical calibration and the latest methods. This is best explained by the direct comparison of the two existing approaches and the proposed method, which we added to the introduction of the revised manuscript.

Reviewer 2: "Also, in the part of "Inertial measurement models", the authors use very ideally assumptions to exclude the impact of the drift of bias and scale factors, the sensor axis mis-alignment, the 1/f noises or other non-Gaussian noise and so on. However, these issues are very common in low-cost MEMS IMU which are widely used in the human body motion detection. So, as a real "plug-and-play" method, the authors should include these non ideal but intrinsic factors into their model and evaluate their impacts to the estimation uncertainty."

This is a good comment, we realize it was not explained how the user of our method should act if their IMUs have significant errors that are not included in our models. In the revised manuscript we have added a new paragraph in Section 2 where these errors, and how they can be handled, are discussed.

For bias drifts, there are other methods that one could use alongside our method to continuously compensate for bias drift. However, even if biases are not compensated for, our method is robust to sensor bias to a certain degree. We show this in a new experiment in the added Section 7.5, where we add randomly generated artificial biases to the measurements. The results of this experiment is shown in Section 8.4 and discussed in Section 9.4.

For other systematic errors, such as scale factors and axis misalignments, we want to point out that we have not compensated for these errors in any way in our experiments and still obtain good results. From our experience, such errors are negligible compared to the biases even for modern low-cost IMUs. However, if such errors are significant, we think it is important to perform a more sophisticated pre-calibration of the sensors before they can be used at all. These errors would not only affect the sensor-to-segment calibration, but the entire motion tracking procedure.

Round 2

Reviewer 2 Report

I have no further commments